# TIAR marks nuclear G2/M transition granules and restricts CDK1 activity under replication stress

Vanesa Lafarga[1,2,3,*], Hsu-Min Sung[1,2,4] (iD), Katharina Haneke[1,2,4], Lea Roessig[1,2], Anne-Laure Pauleau[2,5], Marius Bruer[1,2,4], Sara Rodriguez-Acebes[3], Andres J Lopez-Contreras[3,6], Oliver J Gruss[2,†], Sylvia Erhardt[2,5] (iD), Juan Mendez[3], Oscar Fernandez-Capetillo[3,7] & Georg Stoecklin[1,2,4,**] (iD)

## Abstract

The G2/M checkpoint coordinates DNA replication with mitosis and thereby prevents chromosome segregation in the presence of unreplicated or damaged DNA. Here, we show that the RNA-binding protein TIAR is essential for the G2/M checkpoint and that TIAR accumulates in nuclear foci in late G2 and prophase in cells suffering from replication stress. These foci, which we named G2/M transition granules (GMGs), occur at low levels in normally cycling cells and are strongly induced by replication stress. In addition to replication stress response proteins, GMGs contain factors involved in RNA metabolism as well as CDK1. Depletion of TIAR accelerates mitotic entry and leads to chromosomal instability in response to replication stress, in a manner that can be alleviated by the concomitant depletion of Cdc25B or inhibition of CDK1. Since TIAR retains CDK1 in GMGs and attenuates CDK1 activity, we propose that the assembly of GMGs may represent a so far unrecognized mechanism that contributes to the activation of the G2/M checkpoint in mammalian cells.

Keywords CDK1; cell cycle; G2/M checkpoint; RNA-binding protein; TIAR
Subject Categories Cell Cycle; DNA Replication, Repair & Recombination

See also: **M Altmeyer** (January 2019)

## Introduction

Entry into mitosis is a tightly controlled process essential for faithful inheritance of the genome. The transition from S- to M-phase is driven by the stepwise activation of CDK1: accumulation, nuclear import, and binding of Cyclin B, followed by phosphorylation by CDK7, and removal of inhibitory phosphates by Cdc25 phosphatases [1,2]. The failure to control mitotic entry leads to segregation of partially unreplicated or unrepaired DNA and results in genomic instability, a hallmark of cancer [3]. The ataxia telangiectasia and Rad3-related protein kinase (ATR) acts as a guardian during the transition from S- to M-phase by ensuring that DNA is fully replicated before cells enter mitosis and partition their chromosomes [4]. Slowing or stalling of replication forks, also termed "replication stress", exposes longer patches of single-stranded (ss) DNA, which causes binding of replication protein A (RPA) and recruitment of ATR [5]. The ATR-Chk1 pathway then activates the G2/M checkpoint through Chk1-mediated phosphorylation of Cdc25B and Cdc25C, thereby preventing the dephosphorylation and activation of CDK1 [6–8]. If the ATR-Chk1 pathway cannot be activated in the presence of stalled replication forks, cells fail to resume DNA synthesis, which in turn causes the collapse of replication forks, the entry of unreplicated DNA into mitosis, and the formation of DNA double-strand breaks (DSBs) [9,10]. Indeed, ATR inhibitors do not cause γH2AX accumulation (as a marker of DSBs) in the absence of Cdc25A [11], demonstrating the essential role of controlling mitotic entry.

Replication fork stalling naturally occurs in the presence of DNA adducts, DNA inter-strand crosslinks, or nucleotide depletion [5]. Forced proliferation by the deregulated expression of oncogenes also generates DNA damage through replication stress and leads to the activation of ATR [12]. Moreover, collisions between the replication and transcription machineries are thought to be a major cause of replication stress [13]. Highly transcribed gene clusters are associated with replication fork stalling and were proposed to form early replicating fragile sites that undergo spontaneous DNA breakage [14]. While it is clear that the ATR-Chk1 pathway is essential to coordinate replication and mitotic entry, we still lack a full understanding of how cells ensure that mitosis occurs only when DNA replication is complete, and monitor conflicts between transcription and replication.

1  German Cancer Research Center (DKFZ), Heidelberg, Germany
2  Center for Molecular Biology of Heidelberg University (ZMBH), DKFZ-ZMBH Alliance, Heidelberg, Germany
3  Spanish National Cancer Research Centre (CNIO), Madrid, Spain
4  Division of Biochemistry, Center for Biomedicine and Medical Technology Mannheim (CBTM), Medical Faculty Mannheim, Heidelberg University, Mannheim, Germany
5  CellNetworks Excellence Cluster, Heidelberg University, Heidelberg, Germany
6  Department of Cellular and Molecular Medicine, Center for Chromosome Stability and Center for Healthy Aging, University of Copenhagen, Copenhagen, Denmark
7  Science for Life Laboratory, Division of Genome Biology, Department of Medical Biochemistry and Biophysics, Karolinska Institutet, Stockholm, Sweden
   *Corresponding author. Tel: +34 917328000; E-mail: vlafarga@cnio.es
   **Corresponding author. Tel: +49 621 383 71444; E-mail: georg.stoecklin@medma.uni-heidelberg.de
   †Present address: Institute for Genetics, University of Bonn, Bonn, Germany

The RNA-binding proteins TIA1 and TIAR (TIAL1) play important roles in cell survival, proliferation, and stress responses [15]. Via their RNA recognition motifs (RRMs), the two proteins preferentially bind to U-, CU-, or AU-rich RNA sequences [16], but also have affinity toward ssDNA [17]. TIA1 and TIAR are nucleo-cytoplasmic shuttling proteins that regulate alternative splicing by recruiting U1 snRNP to weak 5′ splice sites [18,19]. Under adverse conditions such as heat shock or oxidative stress, TIA1 and TIAR accumulate in the cytoplasm where they act as repressors of translation [20,21]. Via their carboxy-terminal glutamine-rich domain (QRD), the two proteins also participate in the assembly of stress granules, cytoplasmic aggregates of stalled translation pre-initiation complexes [15]. In response to DNA damage, TIAR is phosphorylated and releases GADD45 mRNA for efficient translation [22,23]. The homozygous deletion of TIAR in mice causes embryonic lethality and defective germ cell maturation [24]. In some cell culture models, depletion of TIAR was found to enhance cell proliferation [25–27], whereas knockout (ko) of TIAR was shown to reduce proliferation in mouse embryonic fibroblasts (MEFs) [28]. Moreover, the double ko of TIA1 and TIAR was recently shown to cause a cell cycle arrest, severe mitotic abnormalities, and a translational stress response in HEK293 cells [29]. However, the mechanism by which TIAR controls cell proliferation remains unknown.

Here, we describe an essential function for TIAR in timing mitotic entry and arresting cells at the G2/M boundary when replication is compromised. Under conditions of replication stress, TIAR relocalizes to G2/M transition granules (GMGs), nuclear foci that contain stalled replication forks together with components of the transcription and splicing machinery. TIAR specifically retains CDK1 in GMGs and attenuates the mitotic activity of CDK1, representing a novel mechanism to control the transition from S- to M-phase.

# Results

## TIAR prevents premature mitotic entry

Reduced levels of TIAR are known to promote tumorigenesis, and knockdown (kd) of TIAR enhances proliferation in K562 and HeLa cells [25,27]. However, TIAR knockout MEFs and HEK293 cells lacking both TIA1 and TIAR proliferate more slowly compared to corresponding controls [28,29], suggesting that TIAR has adverse direct and indirect effects on proliferation. To explore the role of TIAR during cell cycle progression more precisely, we transfected HeLa cells expressing H2B-mCherry and EGFP-α-tubulin (HeLa-H2B/tub) [30] with control or TIAR siRNAs, and synchronized them in early S-phase by double thymidine (TT) block. Upon release from the block, the number of mitotic cells was counted by fluorescence microscopy using H2B condensation and tubulin localization as indicators. Interestingly, we observed an earlier increase in mitotic cells in TIAR kd cells (Fig 1A). We further measured the time between S- and M-phase in HeLa-H2B/tub cells by time-lapse microscopy following release from a TT block and found that TIAR kd cells require approximately 1 h less to reach cell division (Fig 1B). Moreover, TIAR kd in unsynchronized cells induced a threefold increase in histone 3 serine 10 phosphorylation (p-H3), a marker of mitosis (Fig 1C). TIAR kd did not

systematically alter ethynyl-deoxyuridine (EdU) incorporation, replication fork progression, or the distance between replication origins; only si-TIAR S70 showed slight changes in the fork rate and inter-origin distance, indicating that this might be an off-target effect (Appendix Fig S1A–E). Hence, TIAR does not seem to affect the dynamics of DNA replication. Rather, our results indicate that loss of TIAR accelerates mitotic entry. Indeed, kd of Cdc25B was able to prevent premature mitotic entry caused by TIAR depletion (Fig 1D and Appendix Fig S1F).

To ascertain whether accelerated mitotic entry was a specific consequence of TIAR depletion, we generated HeLa$_{dox}$-YFP-TIARr, a HeLa-TREX cell line expressing doxycyline (dox)-inducible, siRNA S62-resistant TIAR tagged with YFP. Indeed, induction of YFP-TIARr-wt was able to restore low p-H3 levels upon kd of endogenous TIAR (Fig 1E). Interestingly, neither TIAR with its three RRMs mutated (TIARr-RRM123m) nor a deletion mutant lacking the QRD (TIARr-dQRD) were able to rescue p-H3 levels in TIAR kd cells (Fig 1F and G), suggesting that both RNA-binding and the aggregation-prone domain are required for TIAR to exert its mitotic function.

## Loss of TIAR causes DSBs and chromosomal aberrations

Since premature mitotic entry may interfere with the completion of replication, which in turn would affect chromosome segregation [10], we next examined if loss of TIAR leads to chromosomal aberrations. In TIAR-depleted cells, a >2-fold increase in chromatin bridges (Fig 2A and Appendix Fig S2A) and an approximately three-fold increase in mitotic extra centrosomes were observed (Fig 2B and Appendix Fig S2B), both of which are typical consequences of premature mitotic entry. Furthermore, metaphase spreads showed that TIAR kd causes an approximately twofold increase in the number of chromosomal breaks (Fig 2C and D). Moreover, we observed pronounced chromosomal cohesion defects after TIAR kd with fully separated, hypercondensed, and scattered chromatids in ~25% of all spreads, whereas this phenotype was not detected in control cells (Fig 2C and E). Since the cohesion defect in TIAR kd cells was attenuated by dox-inducible expression of YFP-TIARr (Appendix Fig S2C and D), we concluded that scattered chromatids are a specific consequence of TIAR depletion.

To explore whether chromosomal aberrations in TIAR kd cells are due to premature mitotic entry, we codepleted Cdc25B and TIAR. Indeed, scattered chromatids and chromosomal breaks were suppressed by kd of Cdc25B (Fig 2F and G), indicating that premature mitotic entry is the major reason for the chromosomal aberrations observed in TIAR-depleted cells.

## TIAR is required for G2/M checkpoint activation in response to replication stress

To further assess whether TIAR prevents mitotic entry in the presence of unreplicated DNA, we induced mild replication stress by treatment with a low dose (0.4 μM) of aphidicolin (APH), an inhibitor of DNA polymerases. At this concentration, cells are affected in S-phase specifically, and replication is perturbed only partially [31]. Quantification of p-H3-positive cells showed that the G2/M checkpoint was strongly activated after 8 h in control cells, whereas TIAR kd cells activated the checkpoint only partially and entered mitosis prematurely (Fig 3A). Release from a TT block confirmed

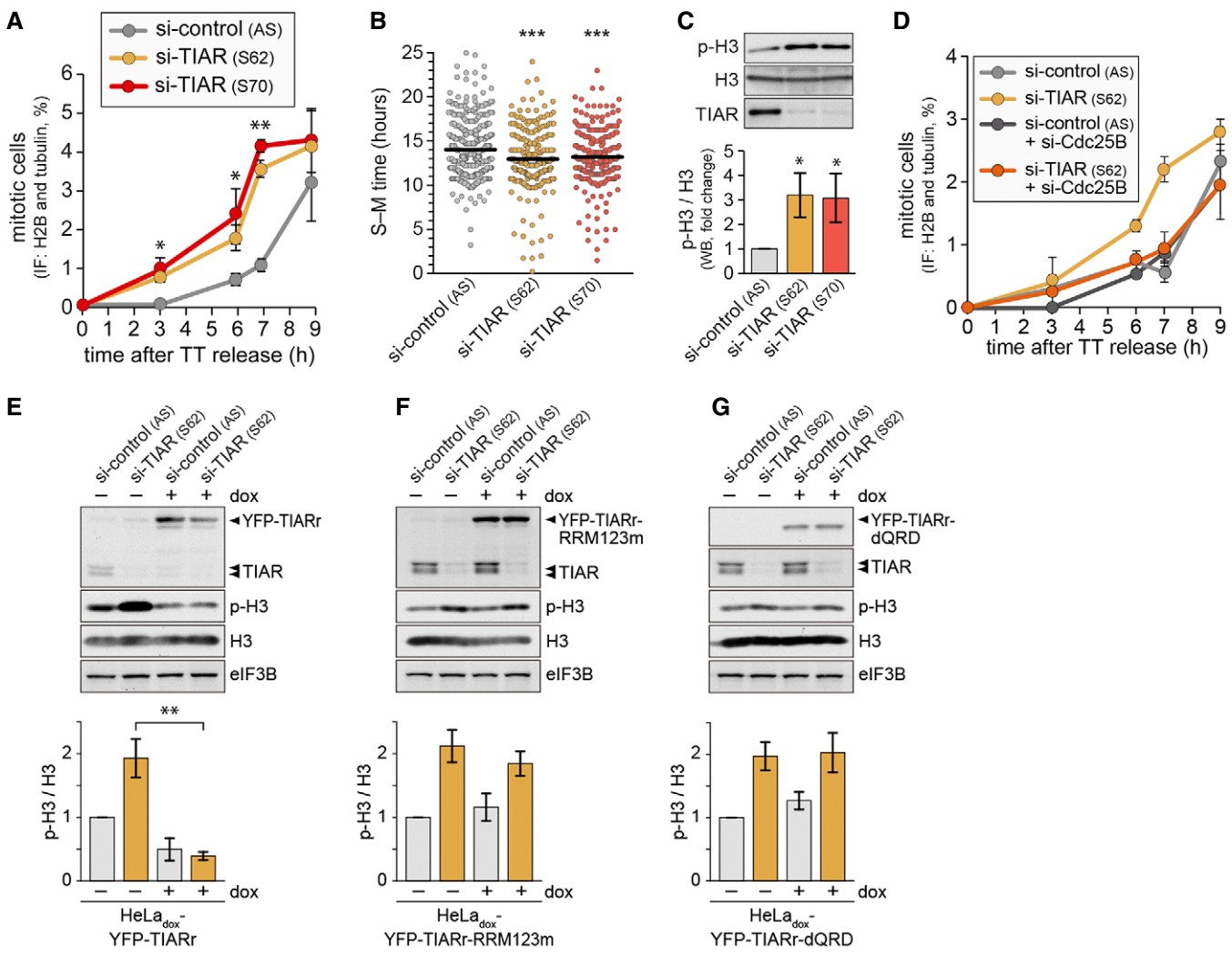

**Figure 1. TIAR controls mitotic entry.**

A  HeLa-H2B/tub cells were transfected with control or TIAR siRNAs for 48 h and synchronized by double thymidine (TT) block. After release from the block, mitotic cells were counted by fluorescence microscopy (mean ± SD; *n* = 3).

B  Time from S- to M-phase was measured in siRNA-transfected HeLa-H2B/tub cells by time-lapse microscopy following release from TT block (4 repeat experiments; all measurements depicted).

C  Western blot analysis of unsynchronized HeLa-H2B/tub cells was carried out to monitor expression of p(S10)-H3, total H3, and TIAR in control and TIAR-depleted cells (mean ± SD; *n* = 3).

D  HeLa cells were transfected with control or TIAR siRNAs, alone or together with Cdc25B siRNA. Seventy-two hours after transfection, p(S10)-H3-positive cells were quantified by flow cytometry (mean ± SD; *n* = 3).

E  HeLa$_{dox}$-YFP-TIARr cells were transfected with control or TIAR siRNAs, and 24 h later, cultured in the absence or presence of 1 μg/ml doxycycline for 48 h. The expression of p(S10)-H3, total H3, and TIAR was measured by Western blot analysis. The graph shows the quantification of p-H3 levels normalized to total H3 (mean ± SD; *n* = 4).

F  HeLa$_{dox}$-YFP-TIARr-RRM123m cells were analyzed as in panel (E) (mean ± SD, *n* = 4).

G  HeLa$_{dox}$-YFP-TIARr-dQRD cells were analyzed as in panel (E) (mean ± SD, *n* = 4).

Data information: Statistical significance was determined by unpaired Student's *t*-test; *$P < 0.05$; **$P < 0.01$; ***$P < 0.001$.

that TIAR depletion causes premature mitotic entry (Appendix Fig S3A). APH treatment reduced the replication fork rate 3.5-fold, an effect that was only slightly weaker in TIAR-depleted cells (three-fold; Appendix Fig S3B). APH shortened the inter-origin distance to the same extent (threefold) in control and TIAR kd cells (Appendix Fig S3C). As a different means to induce replication stress, we also tested a selective ATR inhibitor, ATRi (ETP-46464), which does not inhibit DNAPK or ATM *in cellulo* [9]. ATRi affected

the replication fork rate and reduced the inter-origin distance to a similar degree in control and TIAR kd cells (Appendix Fig S3D and E).

By high-throughput microscopy (HTM), we then assessed the pan-nuclear γH2AX staining as an indicator of replication stress [32,33] (whereas the focal staining is an indicator of DSBs). While TIAR kd did not alter the pan-nuclear γH2AX signal in untreated cells, it strongly increased the signal in APH-treated cells (Fig 3B).

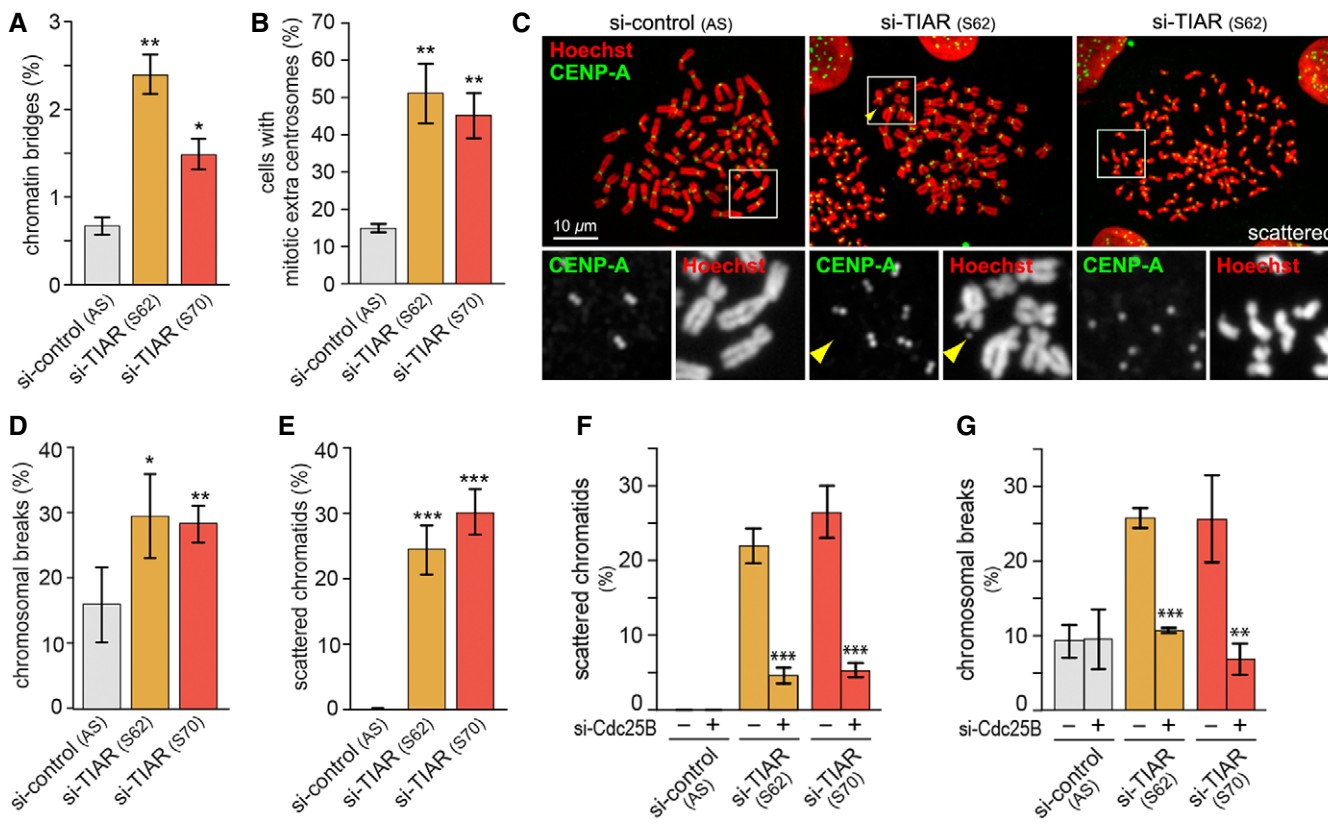

**Figure 2. TIAR deficiency induces chromosomal instability.**

A    Following transfection of HeLa cells with control or TIAR siRNAs for 72 h, mitotic cells were analyzed by IF microscopy and scored for chromatin bridges (mean ± SEM, *n* = 3).

B    HeLa cells were analyzed as in (A) and scored for spindle defects (mean ± SEM, *n* = 3).

C    HeLa cells were transfected with control or TIAR siRNAs for 4 days and subjected to colcemid for 2 h prior to preparation of metaphase spreads. Chromosomes were stained with Hoechst (red) and anti-CENP-A antibody (green); the yellow arrowhead marks a chromosome break.

D    Metaphase spreads were prepared as in (C), and the frequency of cells with chromosomal breaks was quantified (mean ± SD, *n* = 3 independent experiments, approximately 30 metaphase spreads were assessed per experiment and condition).

E    Metaphase spreads were prepared as in (C), and the frequency of cells with scattered chromatids was quantified (mean ± SD, *n* = 3).

F    HeLa cells were transfected with control or TIAR siRNAs, alone or together with Cdc25B siRNA, and 4 days later subjected to colcemid for 2 h prior to preparation of metaphase spreads. The frequency of cells with scattered chromatids was quantified (mean ± SD, *n* = 3 independent experiments, approximately 30 metaphase spreads were assessed per experiment and condition).

G    Metaphase spreads were prepared as in (F), and the frequency of cells with chromosomal breaks was quantified (mean ± SD, *n* = 3).

Data information: In (A, B and D–G), statistical significance was determined by unpaired Student's *t*-test; *$P < 0.05$; **$P < 0.01$; ***$P < 0.001$.

Replication stress induces the formation of 53BP1 nuclear foci [34], which are further elevated by kd of TIAR (Appendix Fig S3F). Moreover, replication stress is tightly associated with exposure of ssDNA, which can be detected as an accumulation of the ssDNA-binding protein RPA in chromatin [12,35]. Indeed, HTM analysis revealed significantly higher levels of chromatin-bound RPA2 in APH-treated cells after TIAR kd (Fig 3C).

Similar to APH, we also observed a pronounced synergy between ATR inhibition and TIAR depletion. In the presence of ATRi, TIAR kd led to a strong increase in the pan-nuclear γH2AX and chromatin-bound RPA2 signal (Figs 3D and EV1A). Both of these effects were mitigated by codepletion of Cdc25B (Fig EV1B and C) or by treatment with Ro3306 (Figs 3D and EV1A), a potent inhibitor of CDK1 that arrests cells at the G2/M boundary [36]. Notably, TIAR depletion had the same sensitizing effect to ATRi in RPE1 cells

(Fig EV1D), a hTERT-immortalized retinal epithelial cell line that retains many properties of primary cells. A synchronization experiment showed that TIAR kd caused a continuous increase in γH2AX levels with a delayed kinetics relative to EdU incorporation (Fig EV1E and F). Taking all these results into account, we concluded that TIAR may have a minor role in S-phase progression under certain conditions of replication stress, yet that it has a major function in controlling mitotic entry.

The combination of ATRi and TIAR kd led to a threefold to fourfold increase in apoptotic (sub-G1) cells as compared to ATRi alone (Fig 3E and Appendix Fig S4A). As expected from the abundance of chromosomal aberrations and damaged DNA, TIAR kd cells subjected to ATRi showed a twofold increase in the number of large, multinucleated cells (Appendix Fig S4B), which result from mitotic catastrophe [37]. In the presence of APH, TIAR kd caused a

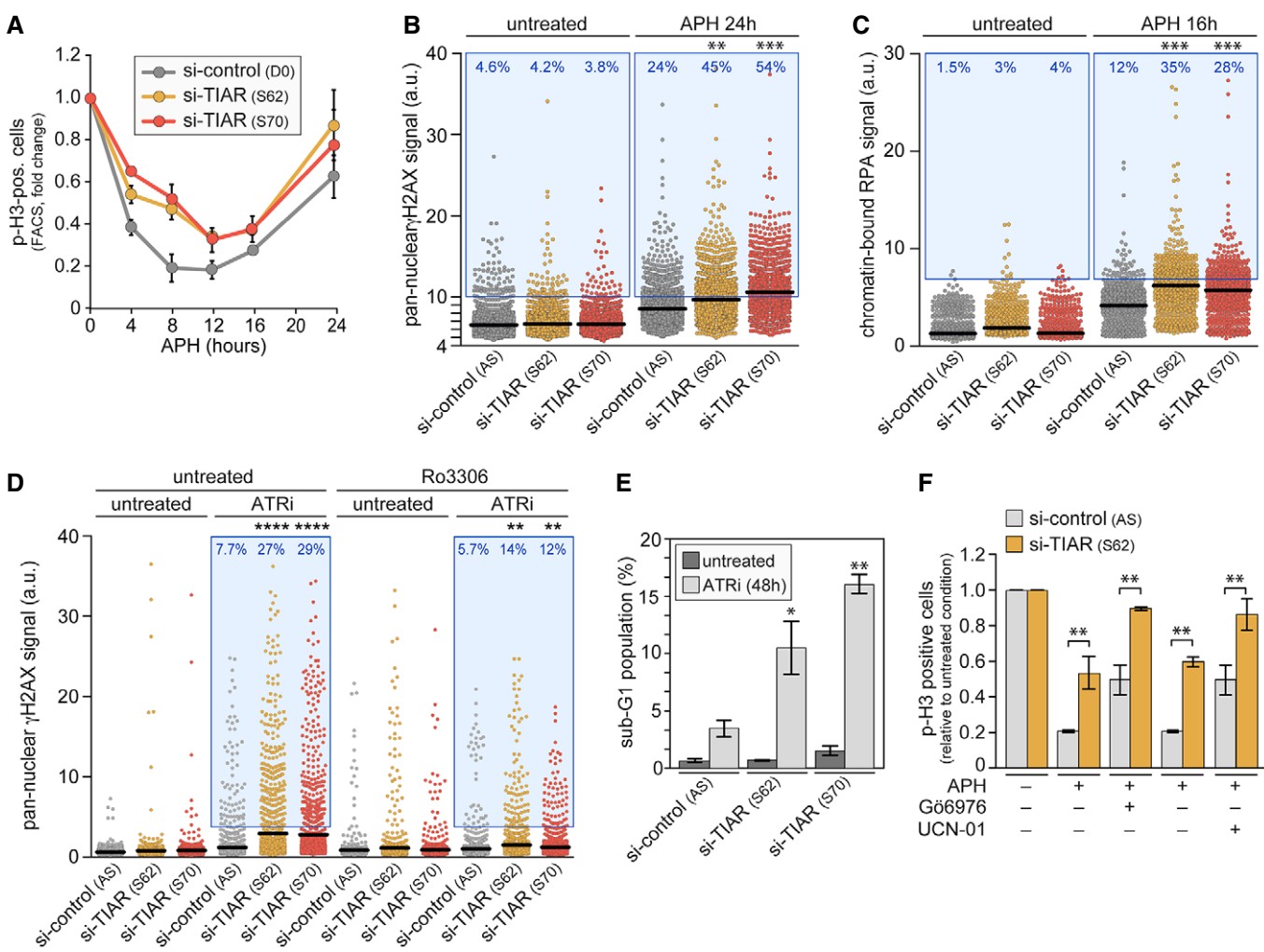

**Figure 3.  TIAR is essential for the replication stress response.**

A   HeLa cells were transfected with control or TIAR siRNAs for 48 h prior to treatment with 0.4 μM APH. Cells were fixed at regular time intervals, and p(S10)-H3-positive cells were quantified by flow cytometry (mean ± SEM, n = 3).

B   HeLa cells transfected with control or TIAR siRNAs were treated with APH for 24 h. Pan-nuclear γH2AX signals were quantified by HTM (n = 3; 2,000 cells examined per experiment and condition). Each dot represents the signal from one cell, horizontal lines indicate mean values, and the blue area delineates cells above an arbitrarily chosen threshold.

C   Chromatin-bound RPA2 signals were quantified by HTM in HeLa cells transfected with control or TIAR siRNAs (n = 3; 600 cells examined per experiment and condition).

D   HeLa cells were transfected with control or TIAR siRNAs for 72 h prior to treatment with ATRi (5 μM) alone or together with Ro3306 (5 μM) for 12 h. Pan-nuclear γH2AX signals were quantified by HTM (n = 3; 1,000 cells examined per experiment and condition).

E   HeLa cells transfected with control or TIAR siRNAs were treated with ATRi for 48 h, and the sub-G1 population was quantified by flow cytometry following propidium iodide staining (mean ± SD, n = 3).

F   HeLa cells transfected with control or TIAR siRNAs were treated with APH, alone or in combination with Gö6976 (1 μg/ml) or UCN-01 (300 nM). After 24 h, p(S10)-H3-positive cells were quantified by flow cytometry and expressed relative to the values in the untreated condition (mean ± SEM, n = 3).

Data information: In (B–D), statistical significance was determined by Wilcoxon rank-sum test. In (E and F), statistical significance was determined by unpaired Student's *t*-test; *P < 0.05; **P < 0.01; ***P < 0.001; ****P < 0.0001.

dramatic, sixfold increase in the number of multinucleated cells (Appendix Fig S4C and D).

Since the ATR-Chk1 pathway is a major inducer of the replication stress response, we tested its contribution to G2/M checkpoint activation with that of TIAR. Inhibition of Chk1 by Gö6976 compromised APH-induced activation of the checkpoint by about 50%, similar to the effect of TIAR kd (Fig 3F). Only the combination of Chk1 inhibition and TIAR kd was able to fully block activation of the checkpoint, indicating that TIAR contributes to G2/M checkpoint activation independently of the ATR-Chk1 axis. The same result was obtained with UCN-01, another Chk1 inhibitor (Fig 3F). These results reveal a strong synergism between the ATR pathway and TIAR, suggesting that premature entry of TIAR-depleted cells into mitosis is due to an impaired replication stress response, causing genome instability, frequent chromosomal aberrations, and mitotic catastrophe.

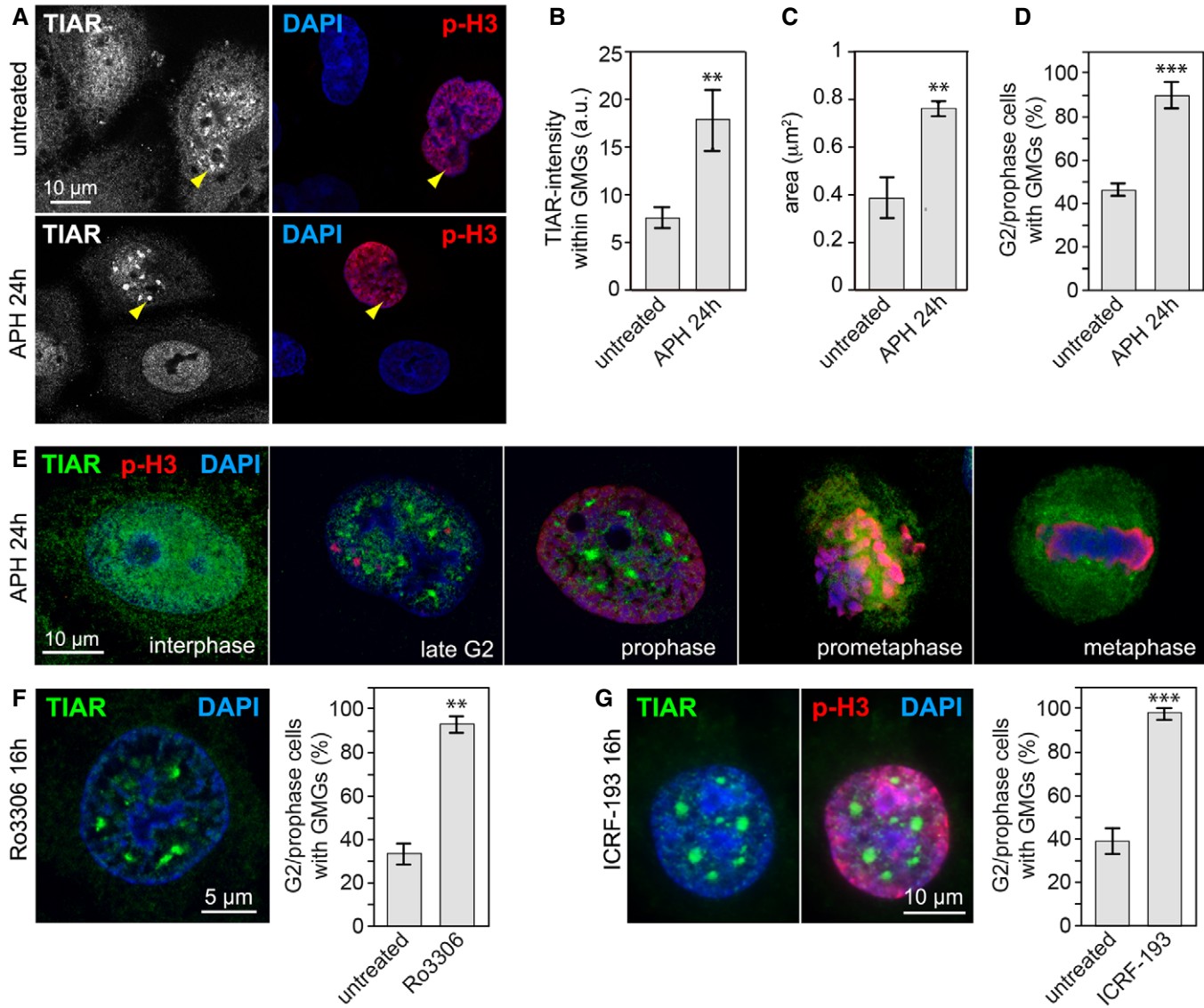

**Figure 4.  Replication stress induces nuclear TIAR containing G2/M transition granules (GMGs) in late G2/prophase.**

A    HeLa cells, untreated or treated for 24 h with 0.4 μM APH, were fixed with methanol and analyzed by IF microscopy after staining with anti-p(S10)-H3 antibody, anti-TIAR antibody, and DAPI. Yellow arrows mark focal accumulation of TIAR in GMGs.

B    IF microscopy was performed as in (A) to determine the intensity of the TIAR signal within GMGs (mean ± SD, *n* = 3 independent experiments, approximately 100 GMGs in 25 cells were analyzed per experiment and condition).

C    IF microscopy was performed as in (A) to determine the area of the TIAR signal within GMGs (mean ± SD, *n* = 3).

D    IF microscopy was performed as in (A) to quantify the percentage of late G2/prophase cells containing GMGs (mean ± SD, *n* = 3 independent experiments, 40 cells were analyzed per experiment and condition).

E    HeLa cells were processed for IF microscopy as in (A), images depict different cell cycle phases.

F, G    HeLa cell were treated for 16 h with (F) the CDK1 inhibitor Ro3306 (9 μM) or (G) the topoisomerase II inhibitor ICRF-193 (1 μM) prior to fixation with methanol and staining with anti-TIAR antibody and DAPI. The graphs show the percentage of late G2/prophase cells containing GMGs (mean ± SD, *n* = 3 independent experiments, 40 cells were analyzed per experiment and condition).

Data information: In (B–D, F and G), statistical significance was determined by unpaired Student's *t*-test; \*\*P < 0.01; \*\*\*P < 0.001.

## TIAR localizes to G2/M transition granules at mitotic entry

To gain further insight into the function of TIAR, we examined its localization during the cell cycle by immunofluorescence (IF) microscopy. While the protein shows a smooth, predominantly nucleoplasmic staining in interphase cells, we noticed that TIAR localizes to discrete nuclear foci specifically when cells start to condense chromatin at the beginning of M-phase (Fig 4A). Upon treatment with APH, the foci became larger and had a more intense TIAR signal, and the frequency of late G2/prophase cells with TIAR foci increased from 46 to 90% (Fig 4B–D). We found that the foci are transient structures that form in a cell cycle-dependent manner

during late G2 and disappear before prometaphase (Fig 4E). Because of their specific occurrence at the G2/M boundary, we termed these foci G2/M transition granules (GMGs).

Specificity of the TIAR staining was confirmed by kd (Appendix Fig S5A), and GMG formation could also be observed with ectopically expressed, YFP-tagged TIAR (Fig EV2A and B). Interestingly, deletion of the glutamine-rich domain prevented TIAR from associating with GMGs (TIARr-dQRD), whereas mutation of the RRMs (TIARr-RRM123m) did not (Fig EV2B).

Similar to TIAR, its paralog TIA1 also showed focal localization in G2/prophase cells, but was less reliable as a marker of GMGs (Fig EV3A and B). However, kd of TIA1 did not cause elevated H3 S10 phosphorylation or impair G2/M checkpoint activation (Fig EV3C–E), nor did it lead to an increase in chromatin breaks or scattered chromatids (Fig EV3F and G). These data suggest that TIAR may have a unique role in G2/M checkpoint activation that is not shared with TIA1.

When cells were treated with the CDK1 inhibitor Ro3306, we observed a similar accumulation of TIAR in GMGs (Fig 4F) concomitant with accumulation of cells in G2/M (Appendix Fig S5B). Accumulation of TIAR in response to Ro3306 is not accompanied by an increase in the $\gamma$H2AX signal (Appendix Fig S5C), indicating that Ro3306 induces TIAR localization due to cell cycle arrest at the G2/M boundary and not by causing replication stress. We further examined RPE1 cells and confirmed that treatment with APH or Ro3306 induces the formation of the same structures (Appendix Fig S5D). In other cell lines such as HCT116 and mouse NIH3T3 fibroblasts, TIAR also localizes to GMGs (Appendix Fig S6A and B). By live microscopy of HeLa$_{dox}$-YFP-TIARr cells, we could verify that YFP-tagged TIAR assembles transiently into nuclear granules (Movie EV1, Appendix Fig S6C). We also observed the granules in cells treated with APH or APH plus Ro3306, where they were more pronounced and persisted longer (Movies EV2 and EV3). In contrast, YFP alone did not assemble into nuclear foci (Movie EV4).

Interestingly, we found that the topoisomerase II inhibitor ICRF-193 is also a highly potent inducer of GMGs (Fig 4G). ICRF-193 is known to prevent chromatid decatenation and induce a G2 arrest without causing DNA strand breaks [38,39]. These results show that GMG formation occurs when cells are arrested at the entry of mitosis, indicating a possible role in G2/M checkpoint activation.

## GMGs contain stalled replication forks, splicing factors, and RNA polymerase II

We then addressed the composition of GMGs and first evaluated components of the replisome. Indeed, PCNA was concentrated in approximately 70% of GMGs (Figs 5A and EV4A), and the ssDNA-binding protein RPA1 showed partial colocalization with GMGs (Fig EV4B). RNase H1, another component of the replication machinery, also accumulates in GMGs (Fig EV4C). FANCD2 was described to localize in small foci corresponding to CFSs [40] and found to be strongly enriched in stalled replication forks [41]. In addition to small foci that presumably correspond to CFSs (red arrow, Fig 5B), we observed colocalization of FANCD2 with TIAR in GMGs (yellow arrows), suggesting the presence of stalled replication forks. Labeling of newly synthesized DNA with a 1-h EdU pulse (Fig EV4D) showed that GMGs do not overlap with sites of late replication, which are frequently associated with CFSs [42].

We then tested whether GMGs correspond to sites of DNA damage. However, DNA damage foci are much smaller, and well-established markers such as 53BP1, $\gamma$H2AX, or Rad51 do not colocalize with TIAR in GMGs (Fig EV4E–G). With BRCA1, we noticed that the small foci do not contain TIAR, whereas large assemblies of BRCA1 colocalize with TIAR in GMGs (Fig 5C). These results show that GMGs are clearly distinct from typical DNA damage foci.

TIAR is known to activate 5′ splice sites [18], and indeed, methylated Sm proteins as well as PRP19, both core components of the spliceosome, perfectly colocalize with TIAR in GMGs (Fig 5D and E). In contrast, Cajal bodies, visualized by staining for Coilin, are smaller in size and number, and showed only a minor accumulation of TIAR (Appendix Fig S7A). Thus, GMGs and Cajal bodies are clearly different structures. HuR, a nuclear RNA-binding protein also involved in splicing [43,44], does not accumulate in GMGs either (Fig 5F), indicating selective recruitment of proteins into GMGs. To determine whether GMGs are associated with transcriptionally active chromatin, we stained for RNA polymerase II (POLR2) using the H5 antibody, which recognizes the elongating POLR2A subunit phosphorylated at S2 of its C-terminal domain. Indeed, pS2-POLR2A colocalizes with TIAR in GMGs (Fig 5G).

TIAR was previously reported to colocalize with the splicing regulator Sam68 and RNA polymerase II in nuclear foci of cells treated with the topoisomerase II inhibitor mitoxantrone [45]. In APH-treated cells, Sam68 foci were separate or adjacent to GMGs, but did not colocalize (Appendix Fig S7B). Moreover, Sam68 foci were observed at all stages of the cell cycle [45], whereas GMGs are specific to late G2 and prophase. From this, we concluded that GMGs are also distinct from Sam68 foci.

Since kd of TIAR did not reduce the focal accumulation of PCNA, RPA1, or PRP19 (Appendix Fig S8A–C), TIAR does not appear to be essential for the formation of GMGs. Taken together, the local concentration of RNA polymerase II, splicing factors, PCNA, RPA1, RNase H1, and FANCD2 indicates that GMGs might be sites of active transcription that assemble around stalled replication complexes during late G2 and prophase.

## Retention of CDK1 in GMGs by TIAR attenuates its kinase activity

Replication stress causes activation of ATR and its effector kinase Chk1, which induces Cdc25 degradation, thereby preventing CDK1 activation and arresting cells in G2 [8]. Since depletion of Cdc25B or treatment with Ro3306 partially rescues TIAR phenotypes (Figs 1D and 2F and G, 3D, and EV1A–C), we explored whether GMGs might be involved in regulating CDK1. Indeed, we observed colocalization of endogenous CDK1 with TIAR in GMGs (Fig 6A), and colocalization could be confirmed with a tagged form of CDK1 (mCherry-CDK1, Fig EV5A). We additionally found Cyclin B to localize in GMGs (Fig 6B), and both CDK1 and Cyclin B were also recruited to GMGs in RPE1 cells (Fig EV5B and C). Importantly, CDK1 was no longer associated with GMGs upon kd of TIAR (Figs 6C and EV5D). Since other GMG components were not affected by TIAR kd (Appendix Fig S8), TIAR causes specific retention of CDK1 in GMGs.

To assess whether TIAR might affect CDK1 activity, the kinase was immunoprecipitated from cellular lysates and tested for its ability to *in vitro* phosphorylate histone H1. This experiment showed that CDK1 is 2.6 times more active when purified from TIAR-depleted cells (Fig 6D), whereas CDK2 activity was not altered

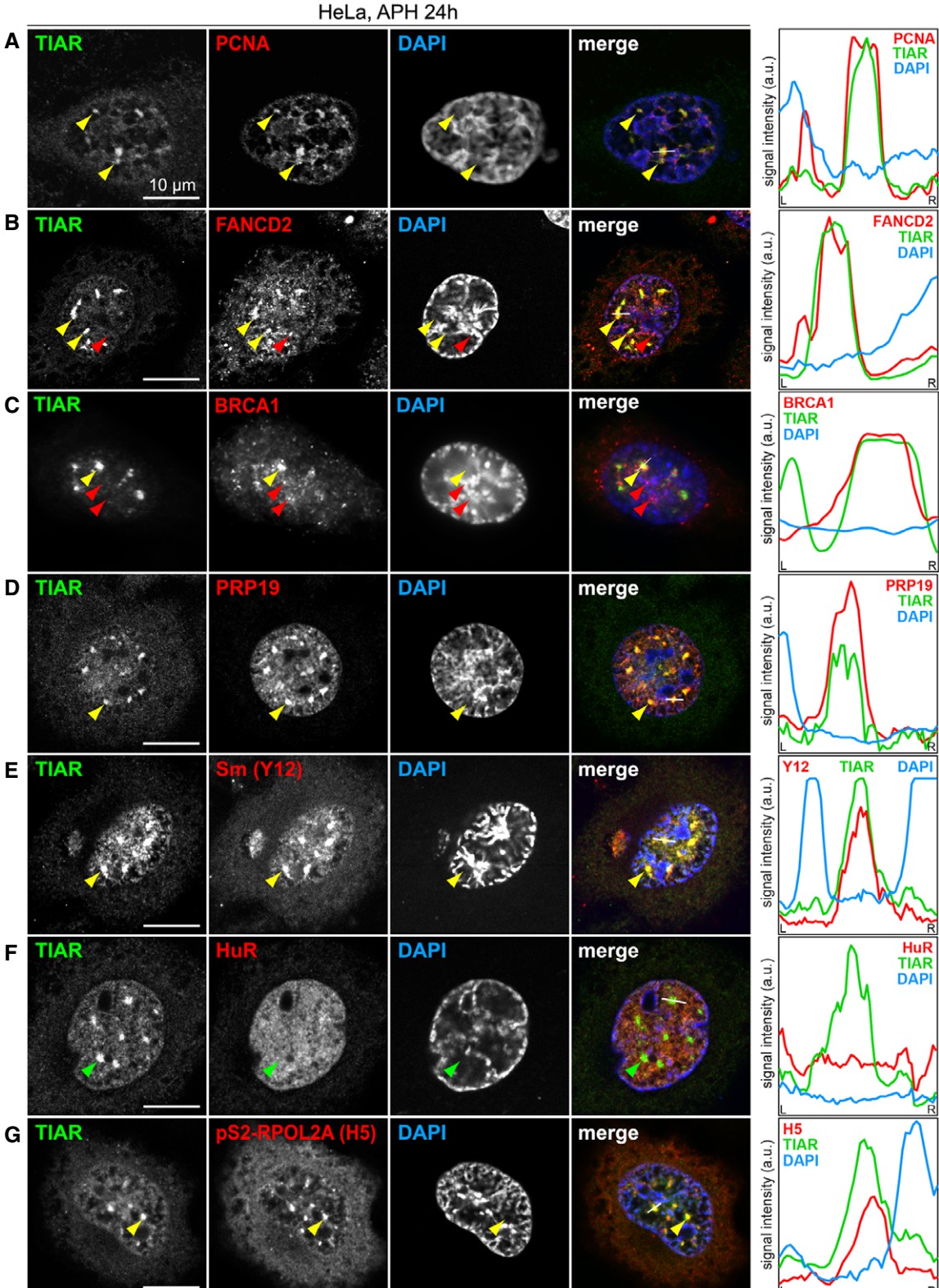

**Figure 5. TIAR colocalizes with transcription and replication components in GMGs.**

A–G   HeLa cells were treated with 0.4 μM APH for 24 h, fixed and processed for IF microscopy after staining with anti-TIAR antibody and DAPI in combination with (A) anti-PCNA antibody, (B) anti-FANCD2 antibody, (C) anti-BRCA1 antibody, (D) anti-PRP19 antibody, (E) anti-Sm(Y12) antibody, (F) anti-HuR antibody, and (G) anti-pS2-RPOL2 (H5) antibody. Intensity profiles along the white line in the merged image are presented on the right side; yellow and green arrows mark GMGs, red arrows mark foci that are distinct from GMGs.

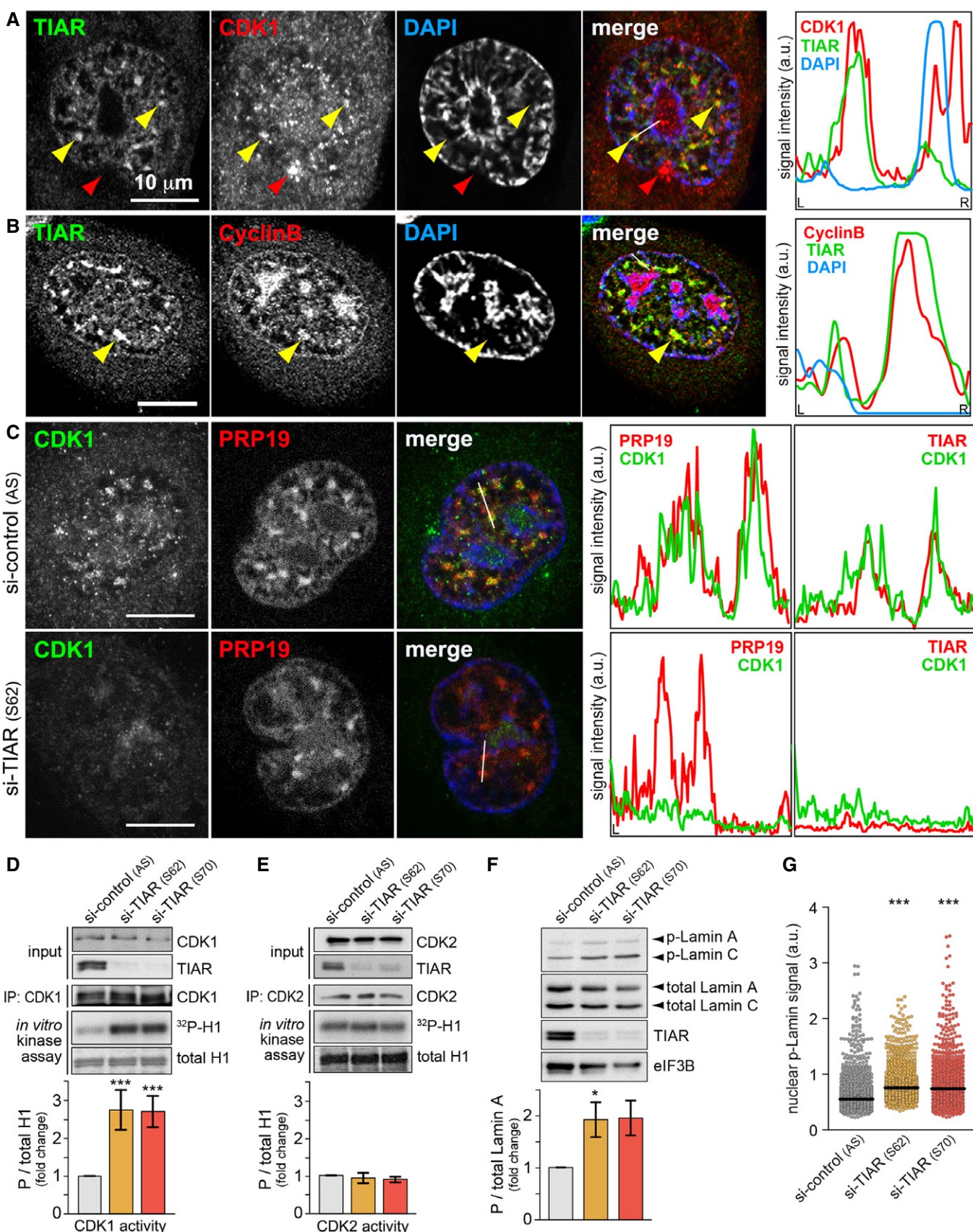

**Figure 6.**

◀

**Figure 6. TIAR retains CDK1 in GMGs and attenuates CDK1 activity.**

A   HeLa cells were treated with 0.4 μM APH for 24 h, fixed, and processed for IF microscopy after staining with anti-TIAR and anti-CDK1 antibodies. Intensity profiles along the white line in the merged image are presented on the right side; yellow arrows mark GMGs, red arrows represent CDK1 that does not colocalize with GMGs.
B   IF microscopy analysis as in (A) using anti-Cyclin B antibody.
C   HeLa cells transfected with control or TIAR siRNAs were treated with 0.4 μM APH 24 h prior to fixation. Cells were processed for IF microscopy after staining with anti-PRP19 and anti-CDK1 antibodies. Intensity profiles from staining of the same cells with anti-TIAR and anti-CDK1 antibodies, shown in Fig EV5D, are depicted on the right side.
D   CDK1 was immunoprecipitated from HeLa cells transfected with control or TIAR siRNAs, tested for *in vitro* kinase activity using recombinant histone H1 as substrate, and visualized by Western blot analysis and autoradiography. The mean CDK1 activity ± SEM was quantified from $n$ = 5 experiments.
E   CDK2 activity was analyzed following immunoprecipitation as in (D) (mean ± SEM, $n$ = 3).
F   HeLa cells were transfected with control or TIAR siRNAs for 72 h before measuring expression levels of p(S22)-Lamin A by Western blot analysis (mean ± SEM, $n$ = 3).
G   p(S22)-Lamin A levels were analyzed by HTM in siRNA-transfected HeLa cells ($n$ = 3; 1,000 cells examined per experiment and condition).

Data information: In (D–F), statistical significance was determined by unpaired Student's *t*-test. In (G), statistical significance was determined by Wilcoxon rank-sum test; *$P$ < 0.05; ***$P$ < 0.001.

(Fig 6E). Accordingly, the phosphorylation level of Lamin A/C, a known target of CDK1 [46], was found to be approximately two times higher in TIAR kd cells as compared to control cells (Fig 6F and G). Importantly, the number of mitotic cells, assessed microscopically by tubulin staining, was elevated only marginally by about 10% after kd of TIAR (Appendix Fig S9A). Hence, elevated CDK1 activity appears to be a cause, and not a consequence, of accelerated mitotic entry in TIAR kd cells. Interestingly, neither CDK1 nor Cyclin B1 levels were affected by kd of TIAR (Appendix Fig S9B–D). Likewise, we did not observe a difference in the phosphorylation status of CDK1 at Y15 or T161 upon kd of TIAR (Appendix Fig S9E and F). Thus, it is conceivable that retention of CDK1 in GMGs by TIAR contributes to the attenuation of CDK1 activity during G2/M checkpoint activation.

## Discussion

This study uncovers a novel and unexpected role for an RNA-binding protein in maintaining genome stability during the normal cell cycle, and in response to replication stress (Fig 7). We propose that TIAR controls CDK1 localization and activity, ensuring proper timing of mitosis. When cells lack TIAR, they enter mitosis prematurely (Fig 1) and show massive defects within mitosis. These include chromosomal breaks, chromatin bridges, mitotic extra centrosomes, and cohesion defects (Fig 2). In addition, we observed pronounced hyperphosphorylation of histone H3 at S10 (Fig 1C), indicating that Aurora B or CDK1 are more active in TIAR-depleted cells. Indeed, this spectrum of phenotypes is typically observed in cells with unscheduled entry into mitosis. Known regulators of CDK1 activity include the inhibitory kinase Wee1 and the activating Cdc25 phosphatases. Cells in which CDK1 is not properly inhibited through Wee1-dependent phosphorylation at Y15 enter mitosis without completing replication, resulting in aberrant mitosis, spindle defects, dispersed chromosomes, and mitotic catastrophe [47–49]. Similarly, when Cdc25B is overexpressed, cells enter prematurely into mitosis and show spindle abnormalities [50,51]. In contrast, depletion of Cdc25B delays mitotic entry and attenuates CDK1-Cyclin B activity [52,53]. Since depletion of Cdc25B in TIAR kd cells prevents premature mitotic entry (Fig 1D) and attenuates the mitotic defects (Fig 2F and G), elevated CDK1 activity (Fig 6D) and unscheduled entry into mitosis are most likely the cause of the mitotic aberrations observed in TIAR-depleted cells. Our results also explain the adverse effects that were observed for TIAR on

proliferation [25,27–29], with loss of TIAR enhancing proliferation through its primary effect of accelerating mitotic entry, yet slowing down proliferation indirectly by causing an accumulation of chromosomal aberrations.

A similar phenotype was previously observed after suppressing the replication stress response through knockout of ATR, which causes cells to enter mitosis prematurely with under-replicated DNA, leading to mitotic DNA breaks and chromatin bridges [4]. Indeed, we found that TIAR is particularly important for activating the G2/M checkpoint upon replication stress (Fig 3A), in line with the pronounced synergism we observed between TIAR kd and ATR inhibition (Figs 3D and E, and EV1A–D). TIAR appears to operate independently of the ATR-Chk1 pathway since Chk1 inhibition and TIAR depletion have an additive effect on suppressing checkpoint activation (Fig 3F). TT release experiments in cells treated with ATRi indicated that TIAR may start to exert its effect in S-phase (Fig EV1E and F). In line with this idea, ATR was recently found to enforce a S/G2 checkpoint by blocking CDK1-directed phosphorylation of the transcription factor FOXM1, which controls a mitotic gene network [54]. We also observed that TIAR kd has a small effect on the replication fork rate in APH-treated cells, although no effect was observed on the inter-origin distance or in response to ATRi (Appendix Fig S3B–E). From this, we concluded that TIAR predominantly acts by controlling mitotic entry, yet it is possible that TIAR starts to exert its effect through an additional mechanism during late S-phase.

Under conditions of APH-induced replication stress, loss of TIAR leads to elevated levels of damaged and ssDNA (Figs 3B–D, EV1A and Appendix Fig S3F). As a consequence, TIAR kd cells experience mitotic catastrophe (Appendix Fig S4B–D), a form of cell death that results from the inability to complete mitosis and is typically associated with the presence of under-replicated DNA [37]. In TIAR kd cells, mitotic catastrophe most likely results from chromosomal breaks and cohesion defects (Fig 2). Taken together, these results demonstrate that TIAR is tightly coupled to the attenuation of CDK1 activity and thereby exerts an essential function in the G2/M transition of normally cycling cells, and acts in late S-phase as well as G2/M checkpoint activation during the replication stress response.

By carefully monitoring the localization of TIAR, we identified GMGs as a subnuclear structure formed in late G2 and prophase (Fig 4). GMGs are induced by replication stress as well as by ICRF-193, a topoisomerase II inhibitor that induces a G2 arrest in an ATR- and BRCA1-dependent manner [39]. Moreover, GMGs are induced by the CDK1 inhibitor Ro3306, reflecting arrest of cells at the G2/M

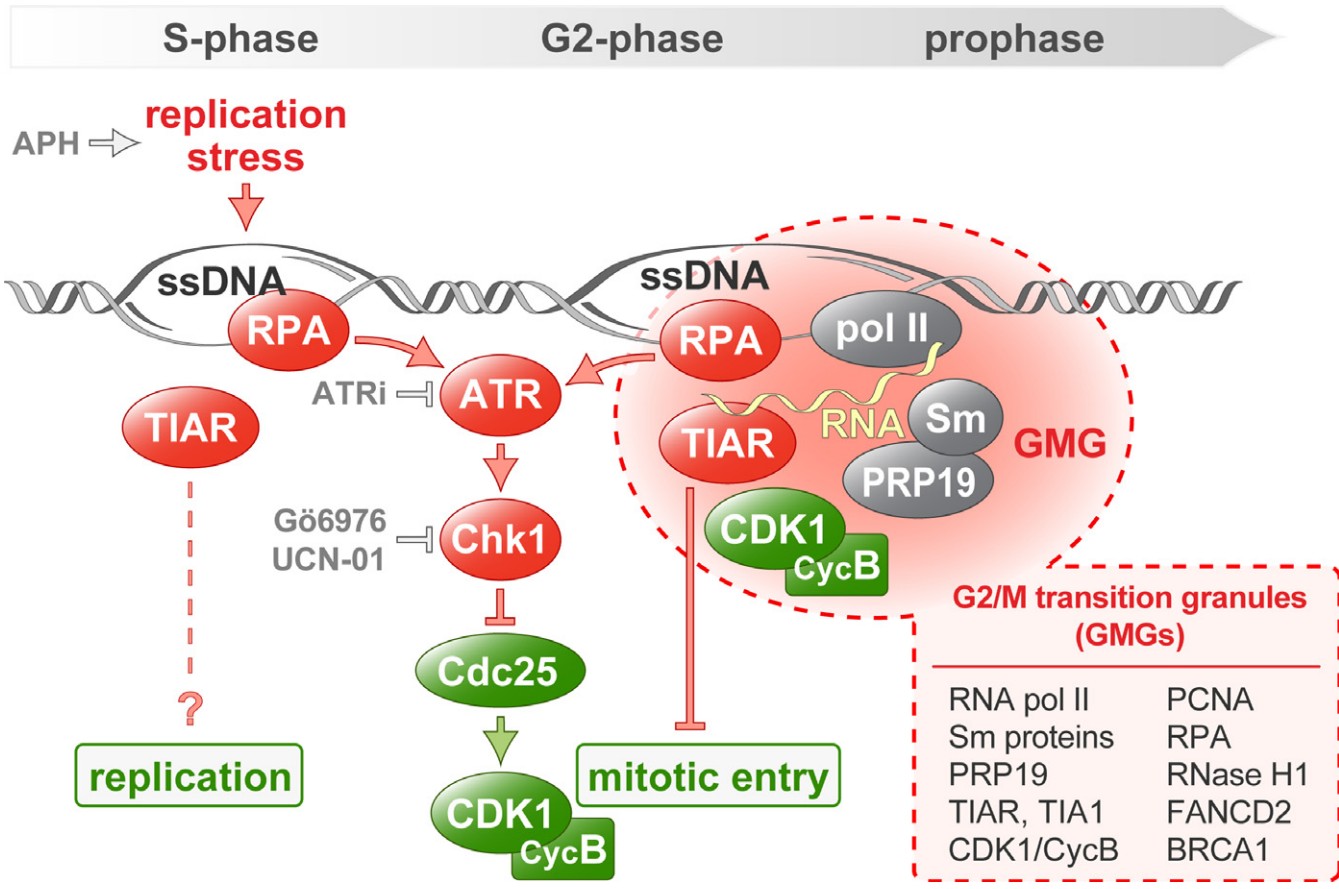

**Figure 7.  Model of TIAR and GMGs in G2/M checkpoint activation.**

The stalling of replication forks is sensed as replication stress and leads to the exposure of ssDNA, which is recognized by RPA. In response to replication stress, the ATR/Chk1 pathway inhibits Cdc25 in order to establish the G2/M checkpoint and prevent mitotic entry. In addition, the formation of GMGs is induced upon replication stress in late G2 and prophase nuclei. GMGs represent assemblies of TIAR together with components of the transcription, splicing, and replication machineries, possibly reflecting active transcription at sites of stalled replication. TIAR retains CDK1 in GMGs and contributes to CDK1 inhibition during G2/M checkpoint activation. APH, DNA polymerase inhibitor aphidicolin; ATR inhibitor ATRi (ETP-46464); Chk1 inhibitors Go6976 and UCN-01 (UCN).

transition and/or a role of CDK1 in the resolution of GMGs. Interestingly, the glutamine-rich unstructured domain of TIAR is required for its localization in GMGs (Fig EV2), which may indicate that GMG formation represents a phase separation event akin to the assembly of intrinsically disordered proteins at sites of DNA damage [55].

GMGs do not contain typical markers of DNA damage (53BP1, γH2AX, or Rad51) and are distinct from the much smaller DNA damage foci (Fig EV4E–G). This fits to our observation that GMGs are formed in response to Ro3306 and ICRF-193 (Fig 4F and G), both of which cause an arrest in G2 without inducing DNA breaks. Recently, the *C. elegans* ortholog of TIAR, TIAR-2, was described to localize in "stress-induced nuclear granules" (SINGs) [56]. Since these granules are formed under conditions of proteotoxic stress, contain proteasome and ubiquitin components, and are not cell cycle phase-specific, we also consider SINGs to be distinct from GMGs.

The accumulation of TIAR, together with S2-phosphorylated RNA polymerase II and core components of the spliceosome (Fig 5D, E and G), suggests that transcription may be active in GMGs. In line with this observation, we found that TIAR requires its RNA recognition motifs to control mitotic entry

(Fig 1F), indicating that it might be associated with nascent RNA during this process. The partial colocalization of BRCA1 with GMGs (Fig 5C) could also point in this direction since BRCA1 is associated not only with DNA repair, but also with transcription regulation [57]. Indeed, BRCA1 was found to occur also outside of DNA damage foci, e.g., in S-phase foci that are associated with replication of pericentric heterochromatin [58] or in nuclear speckles [59].

The presence of PCNA, FANCD2, RPA1, and RNase H1 in GMGs (Figs 5A and B, and EV4B and C), together with the observed induction of GMGs by APH (Fig 4A–D), indicates that stalled replication forks are present in GMGs. Since mitotic chromatin is thought to be transcriptionally silent, it is conceivable that stalled replication forks, where DNA remains single-stranded, cause local activation of transcription, which in turn may serve as a signal to induce the G2/M checkpoint and prevent further progression into mitosis. Alternatively, GMGs may form at sites of conflicts between the replication and transcription machineries, a major source of replication stress leading to replication fork stalling and DSBs [13,14,60]. Moreover, the core splicing factor PRP19, which colocalizes to a high degree with GMGs (Fig 5D),

was previously found to interact with RPA1 at sites of DNA damage, promote activation of ATR, and thereby facilitate recovery of stalled replication forks [61]. Notably, components of the mRNA processing and splicing apparatus represent the largest class of genes that protect cells from genome instability [62]. One plausible model is that RNA synthesis at sites of stalled replication might serve as a signal that contributes to G2/M checkpoint activation.

While depletion of TIAR did not affect the localization of PCNA, RPA, or PRP19 in GMGs (Appendix Fig S8), the central mitotic kinase CDK1 was recruited to GMGs in a TIAR-dependent manner (Figs 6C). Moreover, we observed that Cyclin B also localizes in GMGs (Figs 6B and EV5B). Interestingly, the RRMs of TIAR are required for suppressing mitotic entry (Fig 1F), while they are not essential for accumulation of TIAR in GMGs (Fig EV2). This indicates that an RNA-binding event might be involved in CDK1 attenuation within GMGs. Notably, Cyclin A2 was previously found to have RNA-binding activity [63], raising the possibility that RNA could directly regulate the activity of CDK1 and/or CDK2. Future studies will need to address whether TIAR affects CDK1 through a Cyclin A2-dependent mechanism, and why TIAR attenuates CDK1 but not CDK2 (Fig 6D and E).

It is well established that proper localization of CDK1 is essential for its activation at mitotic entry, and tightly regulated. In G2, CDK1-Cyclin B is kept inactive through inhibitory phosphorylation at T14 and Y15 by the Myt1 and Wee1 kinases. Activation of CDK1-Cyclin B occurs initially at centrosomes, where Cdc25B removes the inhibitory phosphates [52]. Subsequent nuclear import allows activation of the entire pool of CDK1-Cyclin B complexes, which drives the onset of mitosis. Through phosphorylation of numerous target proteins, active CDK1-Cyclin B triggers chromosome condensation, centrosome separation, and assembly of the mitotic spindle, thereby orchestrating progression through pro- and metaphase [64].

Given the central importance of spatial and temporal control of CDK1 activity, retention of CDK1 in GMGs by TIAR may serve as a mechanism to prevent CDK1 from interacting with its mitotic targets. Under conditions of UV irradiation, CDK1 was previously reported to relocalize in nuclear speckles [65]. We consider GMGs distinct from speckles because GMGs contain TIAR, TIA1, PCNA, RPA, RNase H, FANCD2, CDK1, and Cyclin B, all of which are not observed in speckles, and because GMGs are only visible in late G2 and prophase, whereas speckles also occur throughout G1 and S-phase. Nonetheless, both observations suggest that the accumulation of CDK1 in nuclear subcompartments contributes to its inhibition during checkpoint activation. In addition to sequestering CDK1 away from its mitotic targets, our data further suggest that TIAR reduces the activity of CDK1 (Fig 6D). Importantly, TIAR kd did neither affect CDK1 or Cyclin B1 levels, nor the phosphorylation status of CDK1 at Y15 or T161 (Appendix Fig S9). Using recombinant proteins, we could so far not observe a direct interaction between TIAR and CDK1/Cyclin B1, suggesting that a more complex mechanism, which may involve RNA, transcription, and/or splicing, may attenuate CDK1 activity in a TIAR-dependent manner. We propose that GMGs may serve as an inhibitory signaling platform where TIAR together with the transcription and splicing machinery assembles on non-replicated DNA, representing an elegant mechanism to control mitotic kinase activity and timing of the S-M transition as a means to prevent genome instability.

# Materials and Methods

## Cell culture and reagents

HeLa, HCT116, and NIH3T3 cells were maintained in Dulbecco's modified Eagle's medium (DMEM) supplemented with 10% fetal bovine serum (FBS), 2 mM L-glutamine, 100 U/ml penicillin, and 0.1 mg/ml streptomycin at 37°C and 5% $CO_2$. RPE cells were maintained in DMEM:F12 supplemented with 10% fetal bovine serum (FBS), 2 mM L-glutamine, 100 U/ml penicillin, and 0.1 mg/ml streptomycin at 37°C and 5% $CO_2$. For HeLa-H2B/tub cells [30], 1 mg/ml puromycin and 400 μg/ml G418 were added to the medium. For HeLa$_{dox}$-YFP-TIARr cells, 100 μg/ml of zeocin and 5 μg/ml of blasticidin were added to the medium. For synchronization, HeLa and HeLa-H2B/tub cells were subjected to a double thymidine block following standard procedures (18 h 2 mM thymidine, 9 h release, and 18 h 2 mM thymidine). Nocodazole (Sigma) was used at 200 nM, APH (Sigma) at 0.4 μM, UCN-01 (Sigma) at 300 nM, Gö6976 (Calbiochem) at 1 μg/ml, ATRi (ETP-46464 [9]) at 4 μM, and ICRF-193 (Sigma) at 1 μM.

The HeLa$_{dox}$-YFP-TIARr, HeLa$_{dox}$-YFP-TIARr-RRM123m, and HeLa$_{dox}$-YFP-TIARr-dQRD cell lines were generated by transfection of HeLa-TREX cells [66] with pcDNA4/TO-YFP-TIARr (p3380), pcDNA4/TO-YFP-TIARr-RRM123 m (p3414), and pcDNA4/TO-YFP-TIARr-dQRD (p3321), respectively, using Lipofectamine 2000 (Invitrogen). Stably transfected cells were selected with 200 μg/ml of zeocin (Invitrogen) and 10 μg/ml of blasticidin (Invitrogen). After selection, cells were subcloned, and clones #5 (YFP-TIARr), #11 (YFP-TIARr-RRM123 m), and #8 (YFP-TIARr-dQRD) were chosen. Clone #5 (YFP-TIARr) was additionally sorted for YFP-TIARr expressing cells by flow cytometry after adding 1 μg/ml of doxycycline for 18 h.

## siRNA transfection

siRNAs were transfected at a final concentration of 50 nM using Lipofectamine RNAiMAX (Invitrogen) according to the manufacturer's instructions. The following siRNA sequences were used: negative control (AS, AllStars, Qiagen); TIAR (S62) [20] 5′-GGGCUAUUCAUUUGUCAGA-3′; and TIAR (S70) [25] 5′-GUCCUUAUACUUCAGUUGU-3′. S62 and S70 were purchased from Eurofins MWG Operon. Cdc25B siRNA (smart-pool; M-003227-02-0005) was purchased from Dharmacon.

## Live-cell imaging

HeLa-H2B/tub cells [30] expressing H2B-mCherry and EGFP-α-tubulin were kindly provided by Jan Ellenberg (EMBL Heidelberg, Germany) and used for time-lapse microscopy as described previously [67]. Briefly, cells were plated on μ-slide 8-well ibiTreat chambers (Ibidi) and imaged at 5-min intervals for 48 h on a Nikon BioStation IM-Q Time Lapse Imaging System using a 20×/0.8 NA air objective, a 1.3 megapixel cooled monochrome camera (Nikon), and Nikon software for image acquisition. Alternatively, HeLa-H2B/tub cells were imaged at 15-min intervals on a DeltaVision deconvolution microscope using a 20×/0.75 air objective.

HeLa$_{dox}$-YFP-TIARr cells were plated on a 12-well glass bottom plate (Cellvis) and imaged at 15-min intervals for 12 h on a Nikon Ti-E microscope with an integrated perfect focus system using a 40x/0.95 NA air objective, a 4.2 megapixel cooled monochrome sCMOS pco.pge camera, and NIS elements JOBS software for image acquisition. All cells were kept in 10% FBS/DMEM at 37°C and 5% $CO_2$ during imaging. Image processing was performed using ImageJ software (NIH, http://rsbweb.nih.gov/ij/).

## DNA fiber analysis

Exponentially growing cells were pulse-labeled with 50 μM CldU (20 min) followed by 250 μM IdU (20 min). Where indicated, cells were pre-treated with APH or ATRi for 20 min prior to the addition of the CldU and IdU. In the case of TT-synchronized cells, CldU and IdU were added 3 h after release. Thereafter, labeled cells were collected and DNA fibers were spread in buffer containing 0.5% SDS, 200 mM Tris pH 7.4, and 50 mM EDTA. For immunodetection of labeled tracks, fibers were incubated with primary antibodies (for CldU, rat anti-BrdU; for IdU, mouse anti-BrdU) for 1 h at room temperature and developed with the corresponding secondary antibodies for 30 min at room temperature. Mouse anti-ssDNA antibody was used to assess fiber integrity. Slides were examined with a Leica DM6000 B microscope, as described previously [68]. The conversion factor used was 1 μm = 2.59 kb [69]. In each assay, > 300 tracks were measured to estimate fork rate and > 50 fibers containing two or more origins were analyzed to estimate inter-origin distance.

## Western blot analysis

For total protein extracts, cells were washed once with PBS, lysed by directly adding 2× SDS sample buffer, and incubated for 5 min at 95°C. For soluble protein extracts, cells were lysed in RIPA buffer (50 mM Tris–HCl, pH 7.4, 1% NP-40, 0.25% Na-deoxycholate, 150 mM NaCl, and 1 mM EDTA) containing protease inhibitors (cOmplete tablets, Roche) and phosphatase inhibitors (50 mM NaF, 40 nM okadaic acid, and 1 mM Na-vanadate). Samples were resolved by SDS–PAGE, blotted onto 0.2-μm-pore-size nitrocellulose membrane (Peqlab), and blocked in PBS containing 5% milk. HPR-coupled secondary antibodies were purchased from Santa Cruz and Jackson ImmunoResearch, and Western Lightning Plus-ECL Enhanced Luminol Reagent (Perkin Elmer) was used as chemiluminescence substrate.

## Antibodies

Goat anti-TIAR (C-18) (sc-1749), rabbit anti-Sam68 (sc-333), mouse anti-HuR (sc-5261), mouse anti-RNase H1 (sc-136343), mouse anti-CDK1 (Cdc2, sc-137034), mouse anti-CDK2 (sc-163), mouse anti-PCNA (sc-56), rabbit anti-GFP (sc-8334), mouse anti-Cdc25B (sc-1619), rabbit anti-Rad51 (sc-8349), goat anti-TIA1 (sc-1671), and goat anti-eIF3B (eIF3eta) (sc-16377) are from Santa Cruz; rabbit anti-CENP-A (ab13939), mouse anti-actin (sc-47778), rabbit anti-FANCD2 (ab2187), rabbit anti-Lamin-B1 (ab16048), mouse anti-pS2-RPOL2A (H5) (ab24758), rabbit anti-γ-tubulin (ab11317), rabbit anti-H3 (ab1791) and rabbit anti-p-H3 (phospho-Ser10) (ab1136535), and rat anti-BrdU (ab6326) are from Abcam; rabbit anti-RPA1 (2267), rabbit anti-p-Lamin A/C (phospho-Ser22) (13448), rat anti-RPA2 (2208), rabbit anti-p(T161)-Cdk1 (9114), rabbit-p(Y15)-Cdk1 (9111), and mouse anti-Cyclin B (4135) are from Cell Signaling; mouse anti-γH2AX (JBW301) and mouse anti-ssDNA antibody (MAB3034) are from Millipore; rabbit anti-53BP1 (100-304A2) is from Novus Biologicals; mouse anti-α-tubulin (T9026) is from Sigma; mouse anti-BRCA1 (OP92 Ab1) is from Calbiochem; mouse anti-Lamin A/C (477BD7) is from New England Biolabs; and mouse anti-BrdU (BD347580) is from BD Biosciences. Mouse monoclonal anti-Sm (Y12) antibody [70] was kindly provided by Iain Mattaj (EMBL Heidelberg, Germany), rabbit anti-Coilin was kindly provided by Angus I. Lamond (University of Dundee, UK), and rabbit polyclonal anti-PRP19 antibody has been described previously [71].

## Plasmid construction

pEYFP-TIAR (p2181) contains the human TIAR cDNA (long isoform 2) cloned into the *Bgl*II-*Eco*RI sites of plasmid pEYFP-C1 (Clontech) and was kindly provided by Nancy Kedersha and Paul Anderson (Harvard Medical School, Boston, USA). pEYFP-TIARr (p3158) was derived from pEYFP-TIAR (p2181) by site-directed mutagenesis using primers G2845 and G2846 and Quikchange II (Agilent Technologies) according to the manufacturer's instructions. The four silent point mutations in pEYFP-TIARr (p3158) render the mRNA transcribed from this plasmid resistant to the siRNA S62. For pcDNA4/TO-YFP-TIARr (p3380), YFP-TIARr was first amplified by PCR using primers G3288 and G3289, whereby a 5′-*Hind*III and a 3′-*Eco*RI site were created. The PCR product was cloned into the *Hind*III and *Eco*RI sites of plasmid pcDNA4/TO (p2430, Invitrogen) to generate plasmid p3380. pcDNA4/TO-YFP-TIARr-RRM123 m (p3414) was derived from pcDNA4/TO-YFP-TIARr (p3380) by site-directed mutagenesis using primers G3380 and G3381 (RRM1); G3336, G3337, G3348, and G3349 (RRM2); as well as GG3408 and G3409 (RRM3) together with the Quikchange II kit (Agilent Technologies) according to the manufacturer's instructions. pcDNA4/TO-YFP-TIARr-dQRD (p3321) was derived from pcDNA4/TO-YFP-TIARr (p3380) by site-directed mutagenesis using the primers G3352 and G3353 using Quikchange II. mCherry-CDK1 was kindly provided by Ingrid Hoffmann (DKFZ Heidelberg, Germany).

## Primers

G2845, 5′-AAAGGGCTATTCATTCGTGCGGTTTTCAACCCATGAA-3′;
G2846, 5′-TTCATGGGTTGAAAACCGCACGAATGAATAGCCCTTT-3′;
G3288, 5′-GTACAAGCTTATGGTGAGCAAGGGCGAGGA-3′;
G3289, 5′-GATCGAATTCTCACTGTGTTTGGTAACTTGCC-3′;
G3380, 5′-ACAAGCAATGACCCAGCTTGCGCTGTGGAATTTTATGA-3′;
G3381, 5′-TCATAAAATTCCACAGCGCAAGCTGGGTCATTGCTTGT-3′;
G3336, 5′-GGAAAATCCAAAGGCGCTGGTGCTGTATCTTTTTATAA-3′;
G3337, 5′-TTATAAAAAGATACAGCACCAGCGCCTTTGGATTTTCC-3′;
G3348, 5′-TTTTTTCTGGCTACTTGGTGTGGTTGCCCAG-3′;
G3349, 5′-AGTTGCCATGTCTTTAACTACCCGGGCATCCGATATTT-3′;
G3408, 5′-TTCCCAGAAAAGGGCGCTTCAGCTGTGCGGTTTTCAAC-3′;
G3409, 5′-GTTGAAAACCGCACAGCTGAAGCGCCCTTTTCTGGGAA-3′;
G3352, 5′-CCTGATATGACTAAATAACTTCCAACAGGTTG-3′;
G3353, 5′-CAACCTGTTGGAAGTTATTTAGTCATATCAGG-3′.

## Cell cycle analysis

Cells were fixed in 70% ethanol and resuspended in PBS containing 10 μg/ml propidium iodide and 0.5 mg/ml RNase A. Cell cycle profiles were analyzed using a FACSCanto II flow cytometer (BD Biosciences). For quantification of mitotic cells, cells were stained with propidium iodide (DNA content) and anti-p-H3 antibody (anti-phospho histone H3 (Ser10), Alexa Fluor 647 Conjugate, Millipore).

## Metaphase spreads

For analysis of mitotic chromosomes, $5 \times 10^4$ cells were incubated for 2 h with 3.3 mg/ml colcemid (Sigma). Cells were then resuspended in a hypotonic solution of 0.5% Na-citrate and incubated for 10 min. Cells were spun on positively charged glass slides (Thermo Fisher) in a Shandon 4 Cytospin (163 $g$, 10 min), fixed with 4% para-formaldehyde/PBS for 10 min, and stained with anti-CENP-A, anti-Aurora B, or anti-γH2AX antibodies together with Hoechst dye (Sigma) for IF microscopy.

## HTM analyses

HeLa cells transfected with control or TIAR siRNAs were grown on uCLEAR bottom 96-well plates (Greiner Bio-One). Cells were fixed with 2% para-formaldehyde/PBS for 10 min, and IF staining of γH2AX and 53BP1 was performed using standard procedures. For RPA2 IF, cells were pre-extracted with CSK buffer (10 mM PIPES (pH 6.8), 100 mM NaCl, 300 mM sucrose, 3 mM MgCl$_2$, 1 mM EGTA, and 0.5% Triton X-100) for 6 min prior to fixation, which removes the soluble nuclear pool but leaves the chromatin-bound fraction of RPA2. Images were automatically acquired from each well using an Opera High-Content Screening System (Perkin Elmer). Images were segmented based on the DAPI staining to generate masks matching cell nuclei, from which mean signal intensities were calculated. *P*-values were calculated based on a Wilcoxon rank-sum test in all HTM analyses.

## IF microscopy

Cells were seeded onto glass coverslips and, where indicated, transfected with siRNAs for 72 h. For visualization of GMGs and mCherry-CDK1, HeLa and NIH3T3 cells were fixed in methanol for 3 min. For visualization of GMGs in HCT116 cells, an additional permeabilization step using 5% Triton X-100 for 15 min was added. For visualization of PCNA and FANCD2, cells were fixed for 10 min with methanol:acetone (1:1). Coverslips were mounted onto glass slides using a polyvinyl alcohol-based mounting medium and analyzed by fluorescence microscopy using an LSM 780 laser scanning microscope (Zeiss). For visualization of YFP-TIARr, YFP-TIARr-RRM123 m, and YFP-TIARr-dQRD, cells were fixed with 4% PFA in PHEM buffer containing 0.5% Triton X, as described previously [72].

## EdU incorporation

HeLa cells were incubated with 10 mM 5-ethynyl-2′-deoxyuridine (Invitrogen) for the last hour of APH treatment. Cells were fixed with methanol and processed using the Click-iT EdU immunofluorescence kit (Invitrogen) according to the manufacturer's instructions. Cells were further subjected to IF staining as described above.

## Immunoprecipitation and kinase assay

Cell pellets were frozen in liquid nitrogen, mechanically lysed using the TissueLyser II (Qiagen), and resuspended in lysis buffer (Tris–HCl pH 7.5 (20 mM), NaCl (150 mM), MgCl$_2$ (1.5 mM), DTT (1 mM), and complete protease inhibitors (Roche)). Cell lysates were incubated with CDK1-beads (sc-54 AC, Santa Cruz) or CDK2-beads (sc-163 AC, Santa Cruz) for 3 h at 4°C and washed with washing buffer (Tris–HCl pH 7.5 (20 mM), NaCl (300 mM), MgCl$_2$ (2.5 mM), and DTT (1 mM)). Beads were then incubated in 15 μl of H1 Kinase buffer (80 mM β-glycerol phosphate, 20 mM EGTA, 15 mM MgCl$_2$, 1 mM DTT, 20 mM HEPES, 20 mM HEPES, and 50 μM ATP), 2 μCi of $^{32}$P-γ-ATP (3 mCi/mmol, Hartmann Analytic), and 1 μg of histone H1 (Sigma). The reaction was incubated 20 min at room temperature and stopped by adding 6 μl of 6× sample buffer. Samples were analyzed by SDS–PAGE, stained with Coomassie blue, and visualized by autoradiography.

**Expanded View** for this article is available online.

## Acknowledgements

We would like to thank Nancy Kedersha and Paul Anderson (both Harvard Medical School, Boston, USA) for sharing plasmids and thoughtful comments on the manuscript, Thomas Hofmann (DKFZ, Heidelberg) for sharing ideas and reagents, Jan Ellenberg (EMBL Heidelberg, Germany) for sharing the HeLa-H2B/tub cell line, Iain Mattaj (EMBL Heidelberg, Germany) for sharing the anti-Sm (Y12) antibody, Angus Lamond (University of Dundee, UK) for sharing the anti-Coilin antibody, Ingrid Hoffmann (DKFZ Heidelberg, Germany) for sharing the mCherry-CDK1 plasmid, as well as Guillermo de Carcer (CNIO), Ana Losada (CNIO), and Brian Luke (ZMBH, Heidelberg) for thoughtful comments and critical reading of the manuscript. We also thank Holger Lorenz and Diego Megias from the ZMBH and CNIO imaging core facilities for assistance with microscopy and Monika Langloz from the ZMBH FACS facility for help with flow cytometry. We are grateful to Kathrin Bajak for experimental contributions. This work was supported by a Marie-Curie Intra-European Fellowship (mirnaAGOddr, grant Nr. 300384) to VL, doctoral fellowships from the Heidelberg Biosciences International Graduate School to H-MS and MB, grant Nr. 111219 from the Deutsche Krebshilfe to VL and GS, and grant SFB 1036/TP07 from the Deutsche Forschungsgemeinschaft to GS.

## Author contributions

VL made the key findings and performed most experiments. H-MS generated the HeLa$_{dox}$-YFP-TIARr cell lines and did rescue experiments and cell cycle analysis. LR carried out metaphase spreads with assistance of SE. SR-A and JM carried out DNA fibers experiments. VL, AL-C, and OF-C carried out HTM analysis. OJG, KH, MB, and A-LP contributed to cell cycle analysis. VL and GS designed the study, analyzed the data, and wrote the manuscript.

## Conflict of interest

The authors declare that they have no conflict of interest.

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
