## [Review Process File · EMBO Reports]

TIAR marks nuclear G2/M transition granules and restricts CDK1 activity under replication stress

Vanesa Lafarga, Hsu-Min Sung, Katharina Haneke, Lea Roessig, Anne-Laure Pauleau, Marius Bruer, Sara Rodriguez-Acebes, Andres Lopez-Contreras, Oliver J. Gruss, Sylvia Erhardt, Juan Mendez, Oscar Fernandez-Capetillo and Georg Stoecklin

Review timeline:

Submission date:	4 April 2018
Editorial Decision:	14 May 2018
Revision received:	14 September 2018
Editorial Decision:	22 October 2018
Revision received:	6 November 2018
Accepted:	8 November 2018

Editor: Esther Schnapp

Transaction Report:

1st Editorial Decision

14 May 2018

Thank you for your patience while your manuscript was peer-reviewed at EMBO reports. We have now received the enclosed referee reports as well as referee cross-comments.

As you will see, the referees acknowledge that the findings are potentially interesting and novel. However, they also point out that different aspects of the study should be improved. From the cross-comments it becomes clear that potential effects of S phase should be examined, that a little more insight into the mechanism by which TIAR bodies sequester Cdk1 and are activated should be provided, and that the checkpoint bodies should be distinguished from DNA damage foci.

Given these constructive comments, we would like to invite you to revise your manuscript with the understanding that the referee concerns must be fully addressed and their suggestions taken on board. Please address all referee concerns in a complete point-by-point response. Acceptance of the manuscript will depend on a positive outcome of a second round of review. It is EMBO reports policy to allow a single round of revision only and acceptance or rejection of the manuscript will therefore depend on the completeness of your responses included in the next, final version of the manuscript.

Revised manuscripts should be submitted within three months of a request for revision; they will otherwise be treated as new submissions. Please contact us if a 3-months time frame is not sufficient for the revisions so that we can discuss this further. Given the 7 main figures, I suggest that you layout the manuscript as a full article. We can offer a maximum of 5 EV figures, so you may want to combine some EV figures, or move some to the main manuscript file, or move them to an Appendix file. Please see our guide to authors for more information.

Regarding data quantification, please specify the number "n" for how many independent experiments were performed, the bars and error bars (e.g. SEM, SD) and the test used to calculate p-values in the respective figure legends. This information must be provided in the figure legends. Please also include scale bars in all microscopy images.

We strongly encourage the publication of original source data with the aim of making primary data more accessible and transparent to the reader. The source data will be published in a separate source data file online along with the accepted manuscript and will be linked to the relevant figure. If you would like to use this opportunity, please submit the source data (for example scans of entire gels or blots, data points of graphs in an excel sheet, additional images, etc.) of your key experiments together with the revised manuscript. Please include size markers for scans of entire gels, label the scans with figure and panel number, and send one PDF file per figure.

- a complete author checklist, which you can download from our author guidelines (<http://embor.embopress.org/authorguide#revision>). Please insert page numbers in the checklist to indicate where in the manuscript the requested information can be found. The completed author checklist will also be part of the RPF (see below).
- a letter detailing your responses to the referee comments in Word format (.doc)
- a Microsoft Word file (.doc) of the revised manuscript text
- editable TIFF or EPS-formatted figure files in high resolution. In order to avoid delays later in the process, please read our figure guidelines before preparing your manuscript figures at: http://www.embopress.org/sites/default/files/EMBOPress_Figure_Guidelines_061115.pdf

I look forward to seeing a revised version of your manuscript when it is ready. Please let me know if you have questions or comments regarding the revision.

REFeree REPORTS

Referee #1:

Lafarga et al. characterize a role for the RNA-binding protein TIAR in modulating CDK1 activity, checkpoint establishment and entry into mitosis. Loss of TIAR leads to premature entry into mitosis and causes mitotic defects and DNA damage. Delaying mitotic entry by depletion of Cdc25B partly rescues these phenotypes. The authors go on to show that TIAR localizes to what they term checkpoint bodies, sub-nuclear protein assemblies that form in late G2 and contain components of the transcription and splicing machinery. They also demonstrate that CDK1 localizes to these compartments in a TIAR-dependent manner and propose that TIAR sequesters CDK1 within these bodies to attenuate its activity towards mitotic targets, thereby contributing to proper functioning of the G2/M checkpoint.

I reviewed this manuscript before for a different journal and in general consider it an original and interesting piece of work, which puts forward a novel mechanism to tune CDK1 activity and mitotic entry. The manuscript is coherently developed and well written. Especially the loss-of-function phenotypes seem robust, and the majority of the experiments has been performed and analyzed with great care. Nevertheless, there are a couple of points, which I think deserve additional attention and ought to be addressed before this manuscript should be considered for publication. I outline these points below:

- Some phenotypes such as γ H2AX formation and the single stranded DNA marker RPA seem to depend on the addition of exogenous replication stress (Fig. 3), while other phenotypes, such as mitotic entry, are apparent in unstressed conditions and become more prominent when cells experience additional replication stress, including the enhanced formation of checkpoint bodies. If the sensitivity to replication stress is solely due to defective checkpoint establishment, DNA damage markers should primarily increase upon mitotic entry (rather than in S-phase). Time-course experiments, including early time points after replication stress induction, in conjunction with suitable cell cycle markers may help to clarify this point. Along this line, would replication fork rates and inter origin distances remain unaffected by TIAR knockdown when cells are challenged with low doses of APH or with ATRi? Would the RPA-marked single-stranded DNA be rescued by Cdc25 co-depletion? And can the authors show that fork rates, origin firing, and EdU incorporation are unaffected in TIAR-depleted cells upon release from double thymidine block? Without such data it seems very difficult to exclude that S-phase problems contribute to the observed phenotypes.
- It would be important to know which domains of TIAR are required for its localization to checkpoint bodies. Is this dependent on the prion-related QRD? And which domains are needed for CDK1 recruitment to checkpoint bodies and for checkpoint function? In light of the plethora of articles published in the last couple of years on protein assembly into membrane-less compartments by liquid-liquid phase separation, which often depends on low complexity domains, it would be highly informative to comment on the protein properties required for TIAR and CDK1 assembly into checkpoint bodies and discuss the parallels to other phase separated compartments.
- The siRNA-resistant cell lines should be a nice tool to further substantiate some of the key findings. For instance, can the authors test whether CDK1 localization to checkpoint bodies is rescued by YFP-TIARr and compare this to the TIAR mutants (dQRD and RRM123m)?
- The authors suggest that active RNA synthesis occurs within checkpoint bodies. Is the recruitment of PCNA, RPA1 and PRP19 to these structures transcription-dependent? And would transcriptional inhibition lead to an increase in CDK1 activity and entail a checkpoint defect?
- Would it be possible to show time-lapse movies of checkpoint body formation to complement the experiments performed with fixation/extraction and staining?

Additional points:

- The authors begin their manuscript by introducing the RNA-binding proteins TIA1 and TIAR (TIA1-related), and previous findings suggest a functional relationship between these two proteins (e.g. during stress granule assembly). However, the authors never come back to TIA1 and it thus remains unclear to the reader of the article if the two proteins act together in modulating the G2/M checkpoint, or if this is a function unique to TIAR. It would be helpful if the authors could comment on the relationship between these two proteins for mediating the phenotypes described in this manuscript.
- Was the EdU pulse applied for 2h (as stated in the text and figure legend) or for 1h (as stated in the methods section)? An inconsistency also seems to appear when the authors state that RPA focal accumulation can be used to detect ssDNA, but then go on to state that they measured chromatin-bound RPA. This is confusing, and the authors should re-phrase and refer to articles, in which chromatin-bound RPA rather than foci formation has been used to measure ssDNA.

Referee #2:

In this manuscript Lafarga, et al., describe a role for the RNA binding protein TIAR as a mediator of G2/M checkpoint triggered by replication stress. The authors describe sub-nuclear bodies containing TIAR protein, RNA Polymerase II and core splicing factors as well as replication fork and stalled replication fork proteins that they call Checkpoint bodies (ChBs). They observed that ChBs, which are present primarily in G2 and prophase cells occur at low levels in cycling cells and are increased following replication stress. The authors present compelling evidence that ChBs bodies containing TIAR also co-localize with Cdk1 and Cyclin B. Knockdown of TIAR resulted in premature entry into mitosis, elevated Cdk1 activity. Knockdown of TIAR led to premature entry into mitosis as

evidenced by the phospho-H3, DNA damage (53BP1), chromosomal defects, and multinucleated cells when the checkpoint was activated by DNA replication stress caused by low levels of the polymerase inhibitor aph. The checkpoint defects of cells treated with ATR and Chk1 inhibitors were exacerbated by knockdown of TIAR. In addition depletion of Cdc25B rescued the checkpoint defects following TIAR depletion or ATR inhibition.

The authors propose a model based on their findings that TIAR-containing bodies are checkpoint-signaling centers that prevent premature activation of Cdk1 by sequestering Cdk1 in ChBs and in doing so contribute to the activation of the G2/M checkpoint.

Major points

The model proposed by these studies is an intriguing one. However, the data do not elucidate a mechanism by which TIAR-containing bodies sequester Cdk1. These studies also do not identify the signal that activates and regulates the TIAR-containing bodies. These mechanistic insights would make this manuscript stronger and suitable for publication in EMBO reports.

For example, could the ChBs form in response to DNA polymerase and RNA polymerase collisions? Could localized Cdk1 activity play a role in the resolution of those structures? On the other hand, If the phenotypes associated with TIAR depletion are solely due to Cdk1 activity then, a simple experiment to support this model would be to test whether Cdk1 inhibition rescues the phenotypes from TIAR depletion in the presence of aph?

Another loose end: Inhibition of Cdk1 caused an increase in the formation of ChBs in the absence of damage. The authors suggest that Cdk1 inhibition is synchronizing the cells at a phase at which the ChBs can be visualized.

Could the suppression of damage in TIAR depleted cells with replication stress by knockdown of Cdc25B be explained by a reduction in DNA/RNA polymerase encounters or otherwise reduction of stalled forks due to lower levels of DNA replication? In other words Cdc25B depletion could result in the reduction of the source of ChBs rather than inhibiting a signal downstream of ChBs?

Some of these details need to be addressed in order to solidify the model.

Referee #3:

In this report, Lafarga et al examined the impact of the RNA-binding protein TIAR on mitotic entry in cell line models. The foundation of this work settles on the observation that depletion of TIAR led to an early onset of mitotic markers, accompanied by chromosomal aberrations. The authors presented well-executed characterizations of DNA damage surrogate markers in loss of TIAR cells and found that TIAR colocalize with other DDR factors in a subnuclear compartment which they termed as checkpoint bodies. Mechanistically, they show that TIAR exerts its role by indirectly sequestering CDK1 to attain control of mitotic entry and that the ssDNA binding affinity of TIAR allows the recruitment to the sites of replication stress.

Overall, the data are of high quality, all experiments were adequately controlled. The observation that TIAR loss exacerbates damage accumulation from ATR inhibition is based on a modest change but nevertheless interesting. While the experimental data are of high quality in general, the main conclusion seems to be a narrow interpretation favoring the G2/M function. A noticeable issue is whether the available evidence fully qualifies TIAR as a G2/M checkpoint factor, as outlined in the comments below.

Major comments:

1. Figure 1b, the fact that TIAR kd cells exhibited a shortened S phase duration argues for a role of TIAR in S phase. This also explains why synchronized TIAR kd cells exhibit premature/accelerated onset of mitotic markers (Figure 1a). A hastened S phase likely produces an overload of replication stress-derived damages and increase chromosomal aberrations. Therefore, the main conclusion that TIAR control mitotic entry is not fully justifiable.
2. In my view, the term "checkpoint body" is somewhat premature and not fully distinguishable on its own. This is because the characterized components in this paper overlap largely with the DNA damage foci as it is known to the field. The observed subnuclear pattern appears to be a subset or variation of the damage foci. Given that the observed nuclear speckles are TIAR-containing and appears to be G2 specific, the term "checkpoint body" is unfittingly broad and should be omitted.

3. The G2/M checkpoint is commonly accepted as an ATM/ATR-dependent mechanism that restricts mitotic entry in the presence of DNA damage. The authors used aphidicolin-mediated S phase disruption and ATR inhibitor to induce replication stress. It is not clear if TIAR loss actually leads to a compromised G2/M damage check point, such as under ionizing radiation.

Minor comments:

1. Figure 3F, the mock treated control intends to show the percentage of p-H3 positive cells in si-TIAR (S62) and si-control treated cells. However, it seems to be conflicting with Figure 1a where TAIR (S62) is 3-fold higher than si-control.

Cross-comments from referee 3:

I don't disagree with any of the issues raised by Ref 1 and Ref 2. My concerns are a bit more on the conceptual side, namely the new term of "checkpoint body" and the conclusion on G2/M checkpoint. The former does not seem to be distinguishable from the DNA damage foci as the field recognizes. The latter does not align with the manipulations employed. In my view, it is more than likely that there is a strong element of S phase effect that was not considered.

I believe this paper has good originality and data quality. I think the critiques are addressable in general and would suggest the authors be given the opportunity to revise.

Cross-comments from referee 1:

I agree with referee 3 that the concerns, at least most of them, could be addressed within a reasonable time frame. I consider the manuscript original and conceptually very interesting.

One important aspect is indeed the potential contribution of problems in S-phase for the observed phenotypes, and I agree with referee 3 that the provided data do not exclude this possibility. Several experiments have been suggested to address this point and dissect the relative contributions of S-phase versus G2/M checkpoint functions. Ideally, this should help to avoid over-interpretations and reach a more balanced discussion of the results.

Another important aspect is the mechanism by which TIAR-containing bodies sequester CDK1, and the signals that induce them. While in my opinion a complete dissection of the mechanism may be beyond the scope of this manuscript, referee 2 made some valuable suggestions (major point 1), and also I made a couple of suggestions (point 2).

As for the discrimination between DNA damage foci, such as the ionizing radiation induced foci (IRIF), and checkpoint bodies, perhaps the authors could extend their analyses towards co-stainings (under conditions of checkpoint body formation) with additional DDR markers, e.g. MDC1, BRCA1 and RAD51, to complement the findings based on 53BP1 and gammaH2AX co-staining.

Conversely, one could show that CDK1 and CycB do not (but 53BP1, gammaH2AX, MDC1, BRCA1, RAD51 do) accumulate in IRIF (or other DDR compartments associated with DNA double-strand breaks).

1st Revision - authors' response

14 September 2018

Response to reviewers:

Referee #1:

Lafarga et al. characterize a role for the RNA-binding protein TIAR in modulating CDK1 activity, checkpoint establishment and entry into mitosis. Loss of TIAR leads to premature entry into mitosis and causes mitotic defects and DNA damage. Delaying mitotic entry by depletion of Cdc25B partly rescues these phenotypes. The authors go on to show that TIAR localizes to what they term checkpoint bodies, sub-nuclear protein assemblies that form in late G2 and contain components of the transcription and splicing machinery. They also demonstrate that CDK1 localizes to these compartments in a TIAR-dependent manner and propose that TIAR

sequesters CDK1 within these bodies to attenuate its activity towards mitotic targets, thereby contributing to proper functioning of the G2/M checkpoint.

I reviewed this manuscript before for a different journal and in general consider it an original and interesting piece of work, which puts forward a novel mechanism to tune CDK1 activity and mitotic entry. The manuscript is coherently developed and well written. Especially the loss-of-function

phenotypes seem robust, and the majority of the experiments has been performed and analyzed with great care. Nevertheless, there are a couple of points, which I think deserve additional attention and ought to be addressed before this manuscript should be considered for publication. I outline these points below:

Some phenotypes such as γ H2AX formation and the single stranded DNA marker RPA seem to depend on the addition of exogenous replication stress (Fig. 3), while other phenotypes, such as mitotic entry, are apparent in unstressed conditions and become more prominent when cells experience additional replication stress, including the enhanced formation of checkpoint bodies.

If the sensitivity to replication stress is solely due to defective checkpoint establishment, DNA damage markers should primarily increase upon mitotic entry (rather than in S-phase). Timecourse experiments, including early time points after replication stress induction, in conjunction with suitable cell cycle markers may help to clarify this point.

Response: To address this point, we synchronized HeLa cells in G1 with a double thymidine block (TT) and treated them with ATRi during the release. EdU incorporation shows that replication occurred predominantly within the first 3 hours and was completed by 6 hours (Fig EV1F). In TIAR kd cells, the γ H2AX levels started to increase at 6 hours and increased further until 12 hours, when cells were beyond mitosis (Fig EV1E). This result suggested that the absence of TIAR causes H2AX phosphorylation both in late S-phase and upon entry into mitosis.

In addition, we also carried out new experiments to measure γ H2AX in presence of ATRi and the CDK1 inhibitor Ro3306. The result showed that H2AX phosphorylation was prevented almost entirely by Ro3306 treatment (Fig 3D). This is consistent with the strong reduction we previously observed by Cdc25B kd (Fig EV1B). From these results we concluded that TIAR has its major effect during mitotic entry. Yet the TT release experiment also indicated that TIAR may play a role in late S-phase. Since TIAR kd did not cause any difference in the in the DNA fiber analysis in presence of ATRi (and only a very small difference in presence of APH, see next comment), we cannot conclusively assign a role for TIAR during S-phase.

Along this line, would replication fork rates and inter origin distances remain unaffected by TIAR knockdown when cells are challenged with low doses of APH or with ATRi?

Response: We now performed DNA fiber analysis of cells treated with APH (1 μ M), and found that there is a very small increase in the fork rate upon TIAR kd, but no difference in the interorigin distance (Appendix Fig S3B and S3C). When we treated cells with ATRi, we did not observe any difference in the fork rate or inter-origin distance (Appendix Fig S3D and S3E). With these new results we cannot rule out that TIAR, in addition to controlling mitotic entry, may have a function during S-phase under conditions of replication stress. However, this effect appears to be weak, and does not occur under all conditions of replication stress. We changed the corresponding passages in the text (lines 186-188) and indicated a possible additional role of TIAR during S-phase by a question mark in our revised model (Fig 7).

Would the RPA-marked single-stranded DNA be rescued by Cdc25 co-depletion?

Response: We measured the chromatin-bound RPA2 signal by HTM, and found that Cdc25B kd strongly reduced the RPA2 signal in TIAR kd cells treated with ATRi (new data in Fig EV1C). A partial rescue was also observed with the CDK1 inhibitor Ro3306 (new data in Fig EV1A). These results are in line with our notion that TIAR acts primarily in G2/M checkpoint activation, and not during S-phase.

And can the authors show that fork rates, origin firing, and EdU incorporation are unaffected in TIAR-depleted cells upon release from double thymidine block? Without such data it seems very difficult to exclude that S-phase problems contribute to the observed phenotypes.

Response: To address this question, we conducted EdU incorporation as well as DNA fiber

analysis upon release from a TT block in unstressed cells. The result did not show any systematic difference in the rate of EdU incorporation upon TIAR kd (new data in Appendix Fig S1A). Likewise, there was no systematic difference in the fork rate or the inter-origin distance upon TIAR kd (new data in Appendix Fig S1B and S1C). The S70 TIAR siRNA led to a small reduction in the fork rate as well as to a reduction in the inter-origin distance, presumably as a means to maintain a normal overall replication rate. A reduced inter-origin distance with the S70 TIAR siRNA was also visible in our initial analysis with unsynchronized cells (Appendix Figure S1D). We consider this an off-target effect of the S70 siRNA since it did not occur with the S62 siRNA. Taken together, the new results are consistent with our measurements in unsynchronized cells (Appendix Fig S1D, S1E) and confirm that there is no evidence for TIAR having a strong impact on replication.

It would be important to know which domains of TIAR are required for its localization to checkpoint bodies. Is this dependent on the prion-related QRD? And which domains are needed for CDK1 recruitment to checkpoint bodies and for checkpoint function? In light of the plethora of articles published in the last couple of years on protein assembly into membrane-less compartments by liquid-liquid phase separation, which often depends on low complexity domains, it would be highly informative to comment on the protein properties required for TIAR and CDK1 assembly into checkpoint bodies and discuss the parallels to other phase separated compartments.

Response: We tested the importance of TIAR subdomains for localization to GMGs (G2/M transition granules, the foci we had termed Checkpoint bodies in our previous manuscript) using our HeLa cells stably expressing doxycycline-inducible TIAR WT and mutants. The result showed that the RRMs are not required for GMG localization, whereas the C-terminal QRD is needed (new data in Fig EV2). It is interesting to note that the aggregation-prone QRD of TIA proteins also participates in the assembly of cytoplasmic stress granules, suggesting that formation of GMGs may also represent a phase-separation event. We now mention this in the discussion (line 364-367).

The siRNA-resistant cell lines should be a nice tool to further substantiate some of the key findings. For instance, can the authors test whether CDK1 localization to checkpoint bodies is rescued by YFP-TIARr and compare this to the TIAR mutants (dQRD and RRM123m)?

Response: We agree that the YFP-TIARr cells would ideal to validate our CDK1 localization results. However, this turned out to be technically more challenging than we thought. The first problem is that the fixation protocol we have to use for YFP-TIARr (PHEM buffer) is not compatible with the CDK1 antibody – the staining gives a lot of background. Moreover, we cannot verify the TIAR kd efficiency (as e.g. in Fig EV5D) since three channels are already in use (YFP, PRP19 and CDK1). In the Reviewer Fig R1 [Figures for referees not shown.], we show an example of one such staining. The result suggests that YFP-TIARr restores colocalization of CDK1 with PRP19 and TIAR in GMGs. Since we are not confident enough about the quality of these images, we prefer not to include them in the manuscript.

The stainings were even more difficult when we intended to look at the localization of CDK1 in presence of the YFP-TIARr-dQRD mutant. In this case, the YFP signal is diffuse (Fig EV2B), and hence we lack a marker for GMGs. While PRP19 does colocalize with TIAR in GMGs to a high degree (Fig. EV4A), it is also present in speckles in interphase cells, and thus cannot be used to identify GMGs. These difficulties did not allow us to faithfully assess the colocalization of CDK1 with GMGs in presence of the TIAR mutants with our current tools. Alternative strategies will be needed to address this question in future.

The authors suggest that active RNA synthesis occurs within checkpoint bodies. Is the recruitment of PCNA, RPA1 and PRP19 to these structures transcription-dependent? And would transcriptional inhibition lead to an increase in CDK1 activity and entail a checkpoint defect?

Response: When we tested if ActD would affect gH2AX in ATRi-treated cells, we noticed that ActD alone caused an increase in the pan-nuclear gH2AX signal in control cells, suggesting that global inhibition of transcription itself leads to replication stress (Reviewer Fig R2A, below). Interestingly, kd of TIAR no longer caused elevated gH2AX signals in presence of ActD, indicating that TIAR may act in a transcription-dependent manner. However, we also noticed that ActD alone causes a strong reduction in histone H3 serine 10 phosphorylation, indicative of

a cell cycle arrest before mitosis (Reviewer Fig R2B) [Figures for referees not shown.]. This makes it very difficult to interpret the data and draw any conclusion as to whether G2/M checkpoint activation is transcription dependent.

Moreover, we found that ActD causes export of TIAR to the cytoplasm (Reviewer Fig R2C) and the same was observed when we used DRB as a more specific RNA polymerase II-inhibitor (Reviewer Fig R2D). Thus, the initial effect we reported, i.e. that ActD prevents TIAR from localizing to GMGs (former Fig EV5C), could be a simple consequence of this export effect. Given that ActD alone has such profound and widespread effects, the larger question - as to whether transcription is required for G2/M checkpoint activation - cannot be assessed with this approach. We therefore decided to remove former Fig EV5C, and rephrase our conclusions more carefully (line 376-384). This question will need to be addressed by future analyses with more subtle and specific approaches.

Would it be possible to show time-lapse movies of checkpoint body formation to complement the experiments performed with fixation/extraction and staining?

Response: Using our HeLa cells stably expressing doxycycline-inducible YFP-tagged TIAR, we were able to observe the transient assembly of YFP-TIAR into nuclear granules in unstressed cells (Movie EV1). We also observed the bodies in cells treated with APH or APH plus Ro3306, where the foci were more pronounced and persisted much longer (Movies EV2 and EV3). In contrast, YFP alone did not assemble into nuclear foci (Movie EV4).

Additional points:

The authors begin their manuscript by introducing the RNA-binding proteins TIA1 and TIAR (TIA1-related), and previous findings suggest a functional relationship between these two proteins (e.g. during stress granule assembly). However, the authors never come back to TIA1 and it thus remains unclear to the reader of the article if the two proteins act together in modulating the G2/M checkpoint, or if this is a function unique to TIAR. It would be helpful if the authors could comment on the relationship between these two proteins for mediating the phenotypes described in this manuscript.

Response: We added a new set of data showing that TIA1 is recruited to GMGs, although to a lesser degree than TIAR (Fig EV3A and EV3B). Moreover, kd of TIA1 did not cause elevated H3 S10 phosphorylation or impair G2/M checkpoint activation (Fig EV3C-E), and caused only a very small increase in chromatin breaks and scattered chromatids (Fig EV3F and EV3G). These results show that TIAR has a unique role in G2/M checkpoint activation that is not shared with TIA1.

Was the EdU pulse applied for 2h (as stated in the text and figure legend) or for 1h (as stated in the methods section)? An inconsistency also seems to appear when the authors state that RPA focal accumulation can be used to detect ssDNA, but then go on to state that they measured chromatin-bound RPA. This is confusing, and the authors should re-phrase and refer to articles, in which chromatin-bound RPA rather than foci formation has been used to measure ssDNA.

Response: The pulse was done for 1 hour and we modified the text accordingly (line 255). We also rephrased the text to make clear that chromatin-bound RPA was quantified (line 172-174), and included an additional reference on the use of chromatin-bound RPA for measurement of ssDNA (Namiki & Zou, 2006, PNAS 103:580-5).

Referee #2:

In this manuscript Lafarga, et al., describe a role for the RNA binding protein TIAR as a mediator of G2/M checkpoint triggered by replication stress. The authors describe sub-nuclear bodies containing TIAR protein, RNA Polymerase II and core splicing factors as well as replication fork and stalled replication fork proteins that they call Checkpoint bodies (ChBs). They observed that ChBs, which are present primarily in G2 and prophase cells occur at low levels in cycling cells and are increased following replication stress. The authors present compelling evidence that ChBs bodies containing TIAR also co-localize with Cdk1 and Cyclin B. Knockdown of TIAR resulted in premature entry into mitosis, elevated Cdk1 activity. Knockdown of TIAR led to premature entry into mitosis as evidenced by the phospho-H3, DNA damage (53BP1), chromosomal defects, and multinucleated cells when the checkpoint was activated by

DNA replication stress caused by low levels of the polymerase inhibitor aph. The checkpoint defects of cells treated with ATR and Chk1 inhibitors were exacerbated by knockdown of TIAR. In addition depletion of Cdc25B rescued the checkpoint defects following TIAR depletion or ATR inhibition.

The authors propose a model based on their findings that TIAR-containing bodies are checkpoint-signaling centers that prevent premature activation of Cdk1 by sequestering Cdk1 in ChBs and in doing so contribute to the activation of the G2/M checkpoint.

Major points:

The model proposed by these studies is an intriguing one. However, the data do not elucidate a mechanism by which TIAR-containing bodies sequester Cdk1. These studies also do not identify the signal that activates and regulates the TIAR-containing bodies. These mechanistic insights would make this manuscript stronger and suitable for publication in EMBO reports.

Response: We wish to point out that this is the first description of G2/M transition granules (GMGs, the foci we had termed Checkpoint Bodies in our previous manuscript), and we clearly reveal signals that induce their formation, e.g. replication stress induced by APH (Fig 4D), or topological DNA stress induced by the topoisomerase II-inhibitor ICRF-193 (new data in Fig 4G). Based on our finding that recruitment of CDK1 into GMGs is dependent on TIAR (Fig 6C), we tested extensively whether TIAR might directly interact with CDK1. However, none of the approaches we chose (Co-IP, in vitro binding assays using different preparations of recombinant TIAR and CDK1/CyclinB) showed an interaction that we can be confident about. These studies have delayed the publication of our findings by at least two years, and while we share the desire for more mechanistic insight, we also feel that our manuscript reports two major findings (the importance of TIAR for G2/M checkpoint activation and the discovery of GMGs) that are novel and interesting, and warrant publication at this stage.

For example, could the ChBs form in response to DNA polymerase and RNA polymerase collisions?

Response: This an interesting question that we tried to answer. We tested if ActD would affect H2AX phosphorylation in ATRi-treated cells, with the idea that collisions between the transcription and replication machineries should be reduced by ActD. However, we noticed that ActD alone causes an increase in the pan-nuclear gH2AX signal, suggesting that global inhibition of transcription itself causes replication stress (Reviewer Fig R2A, above). Also, ActD alone caused a massive drop in histone H3 serine 10 phosphorylation (Reviewer Fig R2B), indicative of a cell cycle arrest before mitosis. These observations are clearly not in line with the idea that collisions between the transcription and replication machineries are a major source of replication stress and/or a signal for inducing the G2/M checkpoint.

Moreover, we found that ActD causes export of TIAR to the cytoplasm (Reviewer Fig R2C), and the same was observed when we used DRB as a more specific RNA polymerase II-inhibitor (Reviewer Fig R2D). Thus, the initial effect we reported, i.e. that ActD prevents TIAR from localizing to GMGs (former Fig EV5C), could be a simple consequence of this export effect. Given that ActD alone causes such profound and widespread alterations, it is very difficult to rule out indirect effects and draw conclusions on the importance of transcription from this approach. We therefore decided to remove former Fig EV5C, and rephrase our conclusions regarding transcription-dependent GMG formation more carefully (line 380-386).

In addition, we stained cells for R-loops using the S9.6 antibody (Reviewer Fig R3A) [Figures for referees not shown.]. In the nucleus, we observed strong staining of nucleoli, which is consistent with reports in the literature on abundant R-loop formation in this compartment. GMGs, however, did not show an enrichment of the S9.6 signal. According to the literature, the pronounced focal staining in the cytoplasm may correspond to mitochondria. In conclusion, we have no evidence that GMGs contain R-loop structures. Along the same lines, we also observe that depletion of TIAR does not cause an increase of R-loops as determined by HTM of the nuclear S9.6 signal (Reviewer Fig R3B).

Could localized Cdk1 activity play a role in the resolution of those structures?

Response: We now added microscopy movies from our HeLa cells stably expressing doxycycline-inducible YFP-tagged TIAR, which show the recruitment of TIAR into nuclear

granules in live cells. We found that GMGs are larger and brighter, and persist longer, when Ro3306 is added to cells in addition to APH (compare Movies EV2 and EV3). This observation would be in line with a role of CDK1 in the resolution of GMGs. We now included this idea in our discussion (line 362-364).

On the other hand, if the phenotypes associated with TIAR depletion are solely due to Cdk1 activity then, a simple experiment to support this model would be to test whether Cdk1 inhibition rescues the phenotypes from TIAR depletion in the presence of aph?

Response: We did rescue experiments using the CDK1-inhibitor Ro3306 in the presence of ATRi and not APH, since this allows for shorter incubation times (12 hours) compatible with CDK1 inhibition. The new result in Fig 3D shows that Ro3306 prevented almost entirely the increase of pan-nuclear gH2AX in TIAR kd cells. Ro3306 also lead to a partial rescue of the chromatin-bound RPA2 signal (new result in Fig EV1A). This is consistent with the rescue of gH2AX and RPA2 when Cdc25B was co-depleted together with TIAR (Fig EV1B and new data in EV1C). These experiments helped us to consolidate our findings and we concluded that TIAR has a major role in controlling mitotic entry.

Another loose end: Inhibition of Cdk1 caused an increase in the formation of ChBs in the absence of damage. The authors suggest that Cdk1 inhibition is synchronizing the cells at a phase at which the ChBs can be visualized.

Response: We now included a cell cycle analysis by FACS, showing the strong accumulation of G2/M cells after 16 and 24 hours of Ro3306 treatment (Appendix Fig S5B). This is consistent with Ro3306 causing a strong increase in the % of cells with GMGs (Fig 4F).

We further included new data to show that the topoisomerase II inhibitor ICRF-193 strongly induces the formation of GMGs (Fig 4G). ICRF-193 causes a G2-arrest without inducing DNA damage. This new result also supports the idea that GMGs are formed as a consequence of a cell cycle arrest in G2 or G2/M, independently of DNA damage.

Could the suppression of damage in TIAR depleted cells with replication stress by knockdown of Cdc25B be explained by a reduction in DNA/RNA polymerase encounters or otherwise reduction of stalled forks due to lower levels of DNA replication? In other words Cdc25B depletion could result in the reduction of the source of ChBs rather than inhibiting a signal downstream of ChBs?

Response: As mentioned above, we noticed that treatment with ActD causes an increase in the pan-nuclear gH2AX signal, suggesting that global inhibition of transcription itself causes replication stress (Reviewer Fig R2A, above). Moreover, we noticed that ActD prevents TIAR from localizing to GMGs, whereas RNA pol II still localized to GMGs (former Fig EV5C). Therefore, inhibition of transcription does not prevent GMG formation - the absence of TIAR from GMGs could rather be explained by the export of TIAR into the cytoplasm upon transcription inhibition (Reviewer Fig R2C and R2D, above). Taken together, these results indicate that collisions between the transcription and replication machineries are neither a major source of replication stress nor a signal for inducing GMGs. The absence of R-loops from GMGs and the fact that TIAR depletion does not increase R-loops (Reviewer Fig R3A and R3B, above) points in the same direction.

We addressed the second part of this comment in the context of Ro3306, which - as mentioned above - prevented almost entirely the increase of pan-nuclear gH2AX in TIAR kd cells (new result in Fig 3D). Ro3306 also led to a partial rescue of the chromatin-bound RPA2 signal (new result in Fig EV1A), similar to what we see with Cdc25B co-depletion. As shown below by EdU incorporation, Ro3306 has only a minimal effect on replication at the concentration that we used for the rescue experiments (Reviewer Fig R4) [Figures for referees not shown.]. Hence, reduced replication cannot account for the rescue of gH2AX and chromatin-bound RPA2 levels in TIAR kd cells.

Referee #3:

In this report, Lafarga et al examined the impact of the RNA-binding protein TIAR on mitotic entry in cell line models. The foundation of this work settles on the observation that depletion of TIAR led to an early onset of mitotic markers, accompanied by chromosomal aberrations. The

authors presented well-executed characterizations of DNA damage surrogate markers in loss of TIAR cells and found that TIAR colocalize with other DDR factors in a subnuclear compartment which they termed as checkpoint bodies. Mechanistically, they show that TIAR exerts its role by indirectly sequestering CDK1 to attain control of mitotic entry and that the ssDNA binding affinity of TIAR allows the recruitment to the sites of replication stress.

Overall, the data are of high quality, all experiments were adequately controlled. The observation that TIAR loss exacerbates damage accumulation from ATR inhibition is based on a modest change but nevertheless interesting. While the experimental data are of high quality in general, the main conclusion seems to be a narrow interpretation favoring the G2/M function. A noticeable issue is whether the available evidence fully qualifies TIAR as a G2/M checkpoint factor, as outlined in the comments below.

Major comments:

1. Figure 1b, the fact that TIAR kd cells exhibited a shortened S-phase duration argues for a role of TIAR in S-phase. This also explains why synchronized TIAR kd cells exhibit premature/accelerated onset of mitotic markers (Figure 1a). A hastened S-phase likely produces an overload of replication stress-derived damages and increase chromosomal aberrations. Therefore, the main conclusion that TIAR control mitotic entry is not fully justifiable.

Response: We now performed DNA fiber analysis of cells treated with APH, and found that there is a very small increase in the fork rate upon TIAR kd, but no difference in the inter-origin distance (new data in Appendix Fig S3B and S3C). When we treated cells with ATRi, we did not observe any difference in the fork rate or inter-origin distance (new data in Appendix Fig S3D and S3E). In unstressed cells, we had shown in our initial manuscript already that TIAR kd does not affect the fork rate or the inter-origin distance in unsynchronized cells (Appendix Fig S1D and S1E), and we could now further substantiated this finding in synchronized cells by measuring fork rate and inter-origin distance 4 hours after release from a double thymidine (TT) block (new data in Appendix Fig S1B and S1C).

We also added new EdU incorporation experiments, which did not show a systematic difference in the rate of EdU incorporation by kd of TIAR in unstressed (Appendix Fig S1A) or ATRi-treated cells (Fig EV1F). In the setting of ATR inhibition, replication occurred predominantly within the first 3 hours and was completed by 6 hours. In our parallel measurement of γ H2AX levels, H2AX phosphorylation showed the first increase 6 hours after TT release, and continued to increase until 12 hours, when cells were beyond mitosis (Fig EV1E). This result suggested that the absence of TIAR causes H2AX phosphorylation both in late S-phase and upon entry into mitosis.

Taken together, the extensive new analysis confirms our initial model that the major function of TIAR is to control mitotic entry. However, the data also indicate that TIAR may have an additional function during S-phase under conditions of replication stress. This effect appears to be weak, does not occur in unstressed cells, and does not occur under all conditions of replication stress. Therefore, we changed the corresponding passages in the text (lines 186-188) and indicated a possible additional role of TIAR during S-phase in our revised model (Fig 7).

2. In my view, the term "checkpoint body" is somewhat premature and not fully distinguishable on its own. This is because the characterized components in this paper overlap largely with the DNA damage foci as it is known to the field. The observed subnuclear pattern appears to be a subset or variation of the damage foci. Given that the observed nuclear speckles are TIAR-containing and appears to be G2 specific, the term "checkpoint body" is unfittingly broad and should be omitted.

Response: We agree with the reviewer that the term "Checkpoint Bodies" might have been too suggestive since we can formally not prove that the observed foci are really required for G2/M checkpoint activation. What we know is that the foci occur specifically during the transition from late G2 to early M-phase, and that they are strongly induced by APH, Ro3306 and, as part of our new data, ICRF-193 (Fig 4). Hence, we decided to change the name into "G2/M transition granules" (GMGs), which is a more descriptive term that does not imply a role in checkpoint activation. Moreover, by calling them "granules" we stay in line with the nomenclature of cytosolic "stress granules" and "stress-induced nuclear granules", both of which are distinct from GMGs but contain TIA proteins.

Regarding the DNA damage foci, we now analyzed two additional factors: Rad51 is visible in

numerous small nuclear foci in APH-treated cells, which were clearly distinct from the larger and less numerous GMGs (new data in Fig EV4G). BRCA1 also localizes to numerous small nuclear foci that are distinct from GMGs (new data in Fig 5C). Interestingly, we noticed that some of the large BRCA1 foci do in fact colocalize with GMBs. Since the small foci, which we had also observed with 53BP1 and gH2AX antibodies (Fig EV4E and EV4F), correspond to DNA damage sites and clearly do not overlap with GMGs, we feel safe to say that GMGs are distinct from DNA damage foci. The partial colocalization of BRCA1 with GMGs may be related to the fact that BRCA1, besides its role in DNA repair, is associated with transcription regulating complexes and was found to occur also outside of DNA damage foci (in S-phase BRCA1 foci associated with replication of pericentric heterchromatin as well as in nuclear speckles, lines 376-384).

In addition, we included new data to show that the topoisomerase II inhibitor ICRF-193 strongly induces the formation of GMGs (Fig 4G). ICRF-193 causes a G2-arrest without inducing DNA damage. This new result further supports the idea that GMGs are formed as a consequence of a cell cycle arrest in G2 or G2/M, independently of DNA damage.

3. The G2/M checkpoint is commonly accepted as an ATM/ATR-dependent mechanism that restricts mitotic entry in the presence of DNA damage. The authors used aphidicolin-mediated S-phase disruption and ATR inhibitor to induce replication stress. It is not clear if TIAR loss actually leads to a compromised G2/M damage check point, such as under ionizing radiation.

Response: We found that TIAR is also important for maintenance of the G2/M checkpoint induced by DNA damage, i.e. dsDNA breaks upon g-irradiation (Reviewer Fig R5A and R5B) [Figures for referees not shown.]. Moreover, we observed an increase in GMGs upon g-irradiation, yet the increase was less pronounced (smaller foci and fewer cells) than with APH (Reviewer Fig R5C) or ICRF-193 (Fig 4G). Likewise, we observed CDK1 to colocalize with TIAR in GMGs upon g-irradiation, whereas gH2AX or 53BP1 did not colocalize with GMGs (Reviewer Fig R5D and R5E). These results validate the role of TIAR in G2/M checkpoint activation, and confirm that GMGs are distinct from DNA damage foci also under conditions of DNA damage. While we are happy to share these data with the reviewers, we prefer not to include them in the current manuscript, for two reasons:

1) With 20 figures, our manuscript is already very heavy on data, and we feel that including the DNA damage results would make it even more bulky to read. The DNA damage results would go largely unnoticed in the wealth of other information.

2) Since the induction of GMGs by g-irradiation is less pronounced than with APH or ICRF-193 (a topoisomerase II inhibitor know to induce a G2 arrest without causing DNA damage), it is well possible that induction of GMGs by g-irradiation is an indirect consequence of secondary replication stress rather than a direct consequence of DNA damage or repair. We would like to address this issue more carefully, and present our results with DNA damage in a separate manuscript.

Minor comments:

1. Figure 3F, the mock treated control intends to show the percentage of p-H3 positive cells in si-TIAR (S62) and si-control treated cells. However, it seems to be conflicting with Figure 1a where TAIR (S62) is 3-fold higher than si-control.

Response: The reviewer's observation is correct: p-H3-positive cells are elevated in TIAR kd cells as compared to control cells, both in non-stressed conditions (Fig 1A and 1C) and in APHtreated cells (Fig 3A). In Fig 3F, we normalized all values within each series (si-control and si-TIAR) to the value in the untreated condition, which allows for a better comparison of the relative effect of the inhibitors. We corrected the labeling of the y-axis in Fig 3F and modified the legend to better explain how the data were normalized (lines 706-707).

Cross-comments from referee 3:

I don't disagree with any of the issues raised by Ref 1 and Ref 2. My concerns are a bit more on the conceptual side, namely the new term of "checkpoint body" and the conclusion on G2/M checkpoint. The former does not seem to be distinguishable from the DNA damage foci as the field recognizes. The latter does not align with the manipulations employed. In my view, it is more than likely that there is a strong element of S-phase effect that was not considered.

I believe this paper has good originality and data quality. I think the critiques are addressable in general and would suggest the authors be given the opportunity to revise.

Response: As outlined in detail above, our new data show that TIAR does not have a major function in S-phase, yet TIAR might start to show a weak effect in late S-phase under certain conditions of replication stress. We adapted the corresponding passages in the text (lines 186-188, 343-345, 358) and indicated a possible role of TIAR during S-phase in our revised model (Fig 7).

Cross-comments from referee 1:

I agree with referee 3 that the concerns, at least most of them, could be addressed within a reasonable time frame. I consider the manuscript original and conceptually very interesting. One important aspect is indeed the potential contribution of problems in S-phase for the observed phenotypes, and I agree with referee 3 that the provided data do not exclude this possibility. Several experiments have been suggested to address this point and dissect the relative contributions of S-phase versus G2/M checkpoint functions. Ideally, this should help to avoid over-interpretations and reach a more balanced discussion of the results.

Response: We conducted several new experiments to examine the consequences of TIAR depletion during S-phase:

- 1) We synchronized HeLa cells in G1 with a TT block and treated them with ATRi during the release. As explained above, gH2AX levels started to increase in TIAR kd cells at 6 hours (late S-phase) and increased further until 12 hours, when cells were beyond mitosis (Fig EV1E). This result suggested that the absence of TIAR causes H2AX phosphorylation both in late S-phase and upon entry into mitosis.*
- 2) We performed DNA fiber analysis of cells treated with APH and ATRi. The results showed only a very small increase in the fork rate upon TIAR kd in response to APH, but no difference in the inter-origin distance (Appendix Fig S3B and S3C). In ATRi-treated cells, TIAR kd had no effect at all on the fork rate or the inter-origin distance (Appendix Fig S3D and S3E).*
- 3) We conducted EdU incorporation as well as DNA fiber analysis upon release from a TT block in unstressed cells. The result did not show any difference in the rate of EdU incorporation upon TIAR kd (Appendix Fig S1A), and there was no systematic difference in the fork rate or the inter-origin distance upon TIAR kd (Appendix Fig S1B and S1C).*
- 4) We measured gH2AX levels in presence of ATRi and the CDK1 inhibitor Ro3306. The result shows that H2AX phosphorylation is prevented almost entirely by Ro3306 treatment (Fig 3D). Since TIAR kd did not cause any difference in the in the DNA fiber analysis in untreated cells or in presence of ATRi (and only a very small difference in presence of APH), and since Ro3306 prevents H2AX phosphorylation in TIAR-depleted cells in response to ATRi, we cannot conclusively assign a role for TIAR during S-phase. However, the TT release experiment suggested that TIAR may play a role in late S-phase, we changed the corresponding passages in the text (lines 186-188) and indicated a possible additional role of TIAR during S-phase by a question mark in our revised model (Fig 7).*

Another important aspect is the mechanism by which TIAR-containing bodies sequester CDK1, and the signals that induce them. While in my opinion a complete dissection of the mechanism may be beyond the scope of this manuscript, referee 2 made some valuable suggestions (major point 1), and also I made a couple of suggestions (point 2).

Response: We agree that it would be interesting to know more about the mechanism by which GMGs are formed, and how CDK1-recruitment occurs. At this point, despite considerable efforts, we cannot give more detailed answers to these questions. As mentioned above, we extensively explored whether TIAR directly interacts with CDK1, but could not obtain solid evidence for a direct recruitment mechanism. Given that only a small fraction of TIAR and CDK1 assemble in GMGs, and that GMGs occur only in a very small proportion of cells (those at G2/M), it may not be surprising that these experiments were so far not successful. Likewise, it is well possible that posttranslational modifications are involved in recruitment of proteins to GMGs, making these experiments even more challenging. We hope that the reviewers understand these difficulties and appreciate the novelty of our findings.

Also, we would like to point out that our manuscript contains several results that provide some mechanistic insight. For instance, we can show that GMGs are tightly linked to a G2/M arrest and do not require DNA damage per se. This is best illustrated by the topoisomerase II inhibitor ICRF-193, which strongly induces the formation of GMGs (Fig 4G), and is known to cause a

G2-arrest without inducing DNA damage.

Moreover, using our HeLa cells stably expressing doxycycline-inducible YFP-tagged TIAR, we were able to observe in live cells the transient assembly of YFP-TIAR in cells treated with APH or APH plus Ro3306, where the foci were more pronounced and persisted much longer (Movies EV2 and EV3). This observation would be in line with a role of CDK1 in the resolution of GMGs. In addition, we tested the importance of TIAR subdomains for localization to GMGs, and found that the RRM1s are not required, whereas the C-terminal QRD is needed for GMG localization (Fig EV2). It is interesting to note that the aggregation-prone QRD of TIA proteins also participates in the assembly of cytoplasmic stress granules, a process that was described as cytosolic phase-separation. Based on the involvement of the TIAR QRD, formation of GMGs may also represent a phase-separation event.

As for the discrimination between DNA damage foci, such as the ionizing radiation induced foci (IRIF), and checkpoint bodies, perhaps the authors could extend their analyses towards costainings (under conditions of checkpoint body formation) with additional DDR markers, e.g. MDC1, BRCA1 and RAD51, to complement the findings based on 53BP1 and gammaH2AX costaining.

Conversely, one could show that CDK1 and CycB do not (but 53BP1, gammaH2AX, MDC1, BRCA1, RAD51 do) accumulate in IRIF (or other DDR compartments associated with DNA double-strand breaks).

Response: We included new IF microscopy data using additional markers of DNA damage sites in APH-treated cells. Rad51 was found to localize in numerous small foci, but did not colocalize with TIAR in the larger GMGs (Fig EV4G). With BRCA1, we noticed that the small foci do not contain TIAR, whereas large assemblies of BRCA1 did colocalize with TIAR in GMGs (Fig 5C). These results show that GMGs are clearly distinct from typical DNA damage foci.

As suggested by the reviewer, we also analyzed whether CDK1 and TIAR localize at IRIFs in response to g-irradiation. We observed CDK1 to colocalize with TIAR in GMGs upon g-irradiation, but CDK1 did not colocalize with gammaH2AX or 53BP1 in IRIFs (Reviewer Fig R5D and R5E). These results firmly establish that GMGs are distinct from DNA damage foci also under conditions of DNA damage.

2nd Editorial Decision

22 October 2018

Thank you for the submission of your revised manuscript. We have now received the full set of referee comments that is pasted below. I am happy to tell you that all referees support the publication of your study now and we can therefore in principle accept it.

A few more changes are however needed. Please address all referee comments in the manuscript text and also send us a point-by-point response with the newly revised manuscript.

Figs 4C,D, Fig EV 3C,D, Fig EV 4 F,G are not called out in the manuscript text. Fig EV 7D is called out but there is no Fig EV 7D. All figure panels need to be called out in the correct order.

Please fill in the first part of the author checklist on statistics. Given that you have calculated statistics these questions need to be answered.

The legend for Fig 2 misses "data information".

The legend for Fig EV3 misses "data information" and p-value definition.

The Appendix Figs S2C, S3A, S4B,C need to define the p-values.

Please send us up to 5 Keywords.

What do you think about changing the title to:

TIAR marks nuclear G2/M transition granules and sequesters CDK1 under replication stress

Does this reflect the data? I think it might give a clearer picture of what TIAR and the GMGs do, but the current title is also OK.

EMBO press papers are accompanied online by A) a short (1-2 sentences) summary of the findings and their significance, B) 2-3 bullet points highlighting key results and C) a synopsis image that is 550x200-400 pixels large (the height is variable). You can either show a model or key data in the synopsis image. Please note that text needs to be readable at the final size. Please send us this information along with the revised manuscript.

REFEREE REPORTS

Referee #1:

It seems that the authors have gone a long way to address my concerns and those of the other reviewers, and that they did what was technically possible to further improve the manuscript. The newly added data panels and movies are all informative and helpful to define the role of TIAR and of the newly identified G2/M transition granules. There are arguably still some mechanistic aspects, which could perhaps be addressed in future studies, but overall I think that the authors ought to be congratulated for a solid and data-rich manuscript and I believe that the findings are compelling and conceptually interesting enough to warrant publication.

There are only a few points, which I felt would deserve additional attention prior to publishing the study and which I hope will be helpful:

- The authors show that the RRM1s or TIAR are not required for GMG localization (EV2). On the other hand, the RRM1s are required to delay mitotic entry (Fig. 1F). How do the authors explain this? It would be helpful to discuss this point better in the manuscript.

- I assume that the EdU labeling was done differently in Appendix S1A versus EV1E,F, which is used to claim that most replication takes place after 3h (although the data on log scale actually suggests that quite a lot of EdU incorporation happens between 3h and 6h). If EdU was left on the cells during the duration of the TT release in EV1E,F this should be clarified in the figure legend. If EdU was only given for 1h at the end of the release, I would not understand the authors' conclusion that replication occurred predominantly within the first 3 hours.

- The authors may want to add TIA1 to the GMG scheme in their model, and correct the typo in CDK1 (misspelled as "CKD1") in the revised abstract. They may also want to briefly discuss the recently published work by the Cimprich lab (Saldivar et al. An intrinsic S/G2 checkpoint enforced by ATR. Science 2018).

Referee #2:

Brief summary of findings

- TIAR depletion leads to elevated levels of histone H3 phosphorylation on Serine 10. This effect requires both the RNA-binding and the aggregation-prone domains of TIAR.
- Deficiency in TIAR leads to chromatid scattering and chromosomal aberrations even in the absence of exogenous genotoxic stress.
- TIAR localizes to nuclear structures that are rich in phospho-H3 Ser-10 staining and these structures/granules increase in response to treatment of cells with either the CDK1 inhibitor Ro3306 or ICRF-193. The authors conclude from this that these structures, they term granules, are present during G2 phase as well as prophase.
- The localization of TIAR to these granules requires its aggregation-prone domain but not the RNA-binding domain.
- These granules are not sites of DNA damage or DNA synthesis but contain some DNA damage repair and transcription proteins such as BRCA1, FANCD2 as well as RPA1, PCNA and RNA POLII.

- TIAR knockdown results in loss of the Cdk1 protein from these granules without affecting CDK1 stability, activity or expression.
- Loss of a functionally-related protein to TIAR, called TIA, does not lead to elevated levels of phospho-H3, but does have a small effect on chromatin stability (as shown by chromatid scattering and breaks). Moreover, TIA also localizes to the same granules.

The new data included in the authors' response have provided more information about the nuclear structures that contain TIAR as well as some information on DNA replication dynamics upon depletion of TIAR. Given some additional edits and experiments (below), I think that this study can be accepted for publication in EMBO reports.

More specifically:

- 1 In the cells in which TIAR was knocked down treated with the Cdk1-inhibitor Ro3306 in the presence or absence ATRi, the authors state that Ro3306 blocked almost entirely the increase in pan-nuclear gamma H2AX caused by TIAR kd and ATRi. However, Ro3306 treatment only resulted in partial rescue of pan-nuclear gamma H2AX. The text should be changed accordingly. The same for the effect on chromatin-bound RPA2 signal (Fig EV1A). The rescue effect in EV1A was not very strong.
- 2 The experiments using ATRi should be described and discussed acknowledging the caveat that ATR inhibition impacts both S-phase progression as well as transition through G2/M. Furthermore, the gamma H2AX in cells in which TIAR was knocked down in the presence or absence of ATRi was only partially rescued by Ro3306. The same for chromatin bound RPA. The authors must correct this in the text.
- 3 At the same time it is not clear from their experiments the dose of APH utilized, impacted both S-phase as well as G2/M progression.
- 4 Double thymidine block halts cells in early S-phase and not in G1 as the process inhibits replication elongation. The interpretation of the experiments using cells released from a double-thymidine block should be interpreted under that assumption.
- 5 The authors use gamma H2AX to infer sites of replication stress. I suggest that chromatin-bound RPA as well as immuno-staining with phosphorylated RPA in asynchronous cells in which TIAR has been knocked down is a better way of visualizing sites of replication stress. Comparing the phospho RPA with gamma H2AX staining would make a stronger case that for these are indeed sites of replication stress.
- 6 A DNA damage signal that gives a clear G2/M arrest such as damage from ionizing radiation would make a stronger argument regarding the contribution of TIAR in the control of the G2 to M transition.
- 7 The TIAR siRNA S70 results do not always follow those obtained with the S60 siRNA so there are definitely off-target effects. The authors should make sure that this is mentioned in the manuscript.

Referee #3:

In this revised manuscript, the author adequately addressed my comments with additional experiments and modifications of the corresponding statements. Specifically, the authors addressed my first comment by further examining the S phase impact of TIAR. Result of these experiments suggest that there were no detectable global changes in fork rate or Edu incorporation, thus downplaying the role of TIAR during S phase, under unperturbed or challenged conditions. They included a statement to include a potential role of TIAR in S phase, which is acceptable. However, the argument that "while replication occurred predominantly within the first 3 hours and was completed by 6 hours (Fig EV1E and EV1F)" is not supported by the Edu intensity data in Fig. EV1F since this experiment is not accurate for the determination of S phase progression /duration for two reasons. First, there were no time points shown after 6 hours. Second, the linear range of the fast throughput Edu quantification was not demonstrated. In fact, it is very

rare for most cultured mammalian cells, including HeLa cells used in this experiment, to have "replication occurred predominately within the first 3 hours" after releasing from thymidine block. The authors should omit this statement or provide support for it. On the same token, the phrase "late S phase" in line 350 should simply be S phase since there no temporal data to substantiate "early" or "late" here.

My second comment is addressed by using the term "G2/M transition granule" in place of the "checkpoint body" to describe the TIAR staining pattern. This is a more specific and acceptable term.

My last comment pertained to the disconnection between the S phase nature of the perturbations used to induce the stress and the conclusion biased toward the G2/M transition. The authors used IR damage to measure mitotic entry by phospho-H3. The result seems to indicate a delayed defect in G2/M checkpoint. This is acceptable.

In general, I believe the revision is much improved and the main conclusion is strengthened by the additional evidence and modification of the paper.

2nd Revision - authors' response

6 November 2018

Response to reviewers:

Referee #1:

It seems that the authors have gone a long way to address my concerns and those of the other reviewers, and that they did what was technically possible to further improve the manuscript. The newly added data panels and movies are all informative and helpful to define the role of TIAR and of the newly identified G2/M transition granules. There are arguably still some mechanistic aspects, which could perhaps be addressed in future studies, but overall I think that the authors ought to be congratulated for a solid and data-rich manuscript and I believe that the findings are compelling and conceptually interesting enough to warrant publication.

There are only a few points, which I felt would deserve additional attention prior to publishing the study and which I hope will be helpful:

- The authors show that the RRM1 or TIAR are not required for GMG localization (EV2). On the other hand, the RRM1s are required to delay mitotic entry (Fig. 1F). How do the authors explain this? It would be helpful to discuss this point better in the manuscript.

Response: This is an interesting aspect and we extended our discussion including a reference showing that CyclinA2 binds RNA (line 412–419).

- I assume that the EdU labeling was done differently in Appendix S1A versus EV1E,F, which is used to claim that most replication takes place after 3h (although the data on log scale actually suggests that quite a lot of EdU incorporation happens between 3h and 6h). If EdU was left on the cells during the duration of the TT release in EV1E,F this should be clarified in the figure legend. If EdU was only given for 1h at the end of the release, I would not understand the authors' conclusion that replication occurred predominantly within the first 3 hours.

Response: We appreciate the reviewer's comment and changed our wording accordingly (line 191). It now says "A synchronization experiment showed that TIAR kd caused a continuous increase of gH2AX levels with a delayed kinetics relative to EdU incorporation (Fig EV1E and EV1F)". EdU treatment was done the same way in Figure EV1E, F and Appendix Figure S1A; this is now explained better in the corresponding figure legends (line 819–822).

- The authors may want to add TIA1 to the GMG scheme in their model, and correct the typo in CDK1 (misspelled as "CKD1") in the revised abstract. They may also want to briefly discuss the recently published work by the Cimprich lab (Saldivar et al. An intrinsic S/G2 checkpoint enforced by ATR. Science 2018).

Response: We thank the reviewer for the observation, and corrected the typo in the revised abstract

(line 46). Also, we included *TIA1* in our model (Fig 7) and added a note on the recent article from the Cimprich lab to our discussion (line 351–353).

Referee #2:

Brief summary of findings • TIAR depletion leads to elevated levels of histone H3 phosphorylation on Serine 10. This effect requires both the RNA-binding and the aggregation-prone domains of TIAR. • Deficiency in TIAR leads to chromatid scattering and chromosomal aberrations even in the absence of exogenous genotoxic stress. • TIAR localizes to nuclear structures that are rich in phospho-H3 Ser-10 staining and these structures/granules increase in response to treatment of cells with either the CDK1 inhibitor Ro3306 or ICRF-193. The authors conclude from this that these structures, they term granules, are present during G2 phase as well as prophase. • The localization of TIAR to these granules requires its aggregation-prone domain but not the RNA-binding domain. • These granules are not sites of DNA damage or DNA synthesis but contain some DNA damage repair and transcription proteins such as BRCA1, FANCD2 as well as RPA1, PCNA and RNA POLII. • TIAR knockdown results in loss of the Cdk1 protein from these granules without affecting CDK1 stability, activity or expression. • Loss of a functionally-related protein to TIAR, called TIA, does not lead to elevated levels of phospho-H3, but does have a small effect on chromatin stability (as shown by chromatid scattering and breaks). Moreover, TIA also localizes to the same granules.

The new data included in the authors' response have provided more information about the nuclear structures that contain TIAR as well as some information on DNA replication dynamics upon depletion of TIAR. Given some additional edits and experiments (below), I think that this study can be accepted for publication in EMBO reports.

More specifically:

1 In the cells in which TIAR was knocked down treated with the Cdk1-inhibitor Ro3306 in the presence or absence ATRi, the authors state that Ro3306 blocked almost entirely the increase in pan-nuclear gamma H2AX caused by TIAR kd and ATRi. However, Ro3306 treatment only resulted in partial rescue of pan-nuclear gamma H2AX. The text should be changed accordingly. The same for the effect on chromatin-bound RPA2 signal (Fig EV1A). The rescue effect in EV1A was not very strong.

Response: We are not sure what this comment refers to since we do not state in our manuscript that “Ro3306 blocked almost entirely the increase in pan-nuclear gamma H2AX caused by TIAR kd and ATRi.” We agree that the rescue is partial, which we describe this accurately by using the word “mitigate” on line 186: “Both of these effects were mitigated by co-depletion of Cdc25B (Fig EV1B and EV1C) or by treatment with Ro3306 (Fig 3D and EV1A), a potent inhibitor of CDK1 that arrests cells at the G2/M boundary”. To make this point clear, we also included the word “partially” when describing the rescue on line 295.

2 The experiments using ATRi should be described and discussed acknowledging the caveat that ATR inhibition impacts both S-phase progression as well as transition through G2/M. Furthermore, the gamma H2AX in cells in which TIAR was knocked down in the presence or absence of ATRi was only partially rescued by Ro3306. The same for chromatin bound RPA. The authors must correct this in the text.

Response: We agree with the reviewer that is an important point. We therefore extended our discussion including a recent article from the Cimprich lab (line 351–353) that describes the role of ATR during S-phase in enforcing a S/G2 checkpoint. The accurate description of the partial rescue by Ro3306 is addressed in the paragraph above.

3 At the same time it is not clear from their experiments the dose of APH utilized, impacted both S-phase as well as G2/M progression.

Response: We added a sentence and reference regarding the effect of APH at low concentrations (line 161–162).

4 Double thymidine block halts cells in early S-phase and not in G1 as the process inhibits replication elongation. The interpretation of the experiments using cells released from a double-thymidine block should be interpreted under that assumption.

Response: We corrected this and changed the corresponding passage in the text to specify that

thymidine blocks cells in early S-phase (line 115–116).

5 The authors use gamma H2AX to infer sites of replication stress. I suggest that chromatin-bound RPA as well as immuno-staining with phosphorylated RPA in asynchronous cells in which TIAR has been knocked down is a better way of visualizing sites of replication stress. Comparing the phospho RPA with gamma H2AX staining would make a stronger case that for these are indeed sites of replication stress.

Response: In fact, we show that gH2AX foci do not colocalize with GMGs in APH-treated cells (Fig EV4F). This is in line with the notion that focal gH2AX staining corresponds to sites of DNA damage, and that GMGs are distinct from DNA damage foci. Our interpretation that GMGs represent sites of stalled replication forks is based on the accumulation of PCNA, FANCD2 and RPA1 (Fig 5A, 5B and EV4A).

6 A DNA damage signal that gives a clear G2/M arrest such as damage from ionizing radiation would make a stronger argument regarding the contribution of TIAR in the control of the G2 to M transition.

Response: We already showed in our previous rebuttal letter that TIAR is also important for maintenance of the G2/M checkpoint induced by g-irradiation (Reviewer Fig R5A and R5B, below). Moreover, we observed an increase in GMGs upon g-irradiation, yet the increase was less pronounced (smaller foci and fewer cells) than with APH (Reviewer Fig R5C) or ICRF-193 (Fig 4G). Likewise, we observed CDK1 to colocalize with TIAR in GMGs upon g-irradiation, whereas gH2AX or 53BP1 did not colocalize with GMGs (Reviewer Fig R5D and R5E) [Figures for referees not shown].

Since our manuscript is already very heavy on data with 20 figures, we feel that including the DNA damage results would make it even more bulky to read. The DNA damage results would go largely unnoticed in the wealth of other information, and we plan to publish them on a separate occasion.

7 The TIAR siRNA S70 results do not always follow those obtained with the S60 siRNA so there are definitely off-target effects. The authors should make sure that this is mentioned in the manuscript.

Response: There is no discrepancy between the effects of siRNA S62 and S70 with regard to the effect on mitotic entry (Fig 1), chromosomal aberrations (Fig 2), synergy with ATR (Fig 3) or CDK1 activity (Fig 6). Moreover, the effect on mitotic entry is backed up by a solid rescue experiment (Fig 1E–G). Hence, we have no reason to invoke off-target effects with regard to our major findings on the role of TIAR in G2-M transition.

The only experiments where the two siRNAs showed a discrepancy were the DNA fiber analyses (Appendix Fig S1B, S1C, S1E and S3D). In all these experiments, siRNA S70 showed a difference in the fork rate or inter-origin distance compared to the control siRNA, whereas S62 did not. Hence, we cannot be sure that TIAR really has an effect on replication, since S70 may indeed have an off-target effect during S-phase. This possible off-target effect is now mentioned on line 125–126.

Corresponding Author Name: Georg Stoecklin

Manuscript Number: EMBOR-2018-46224V1